# *Drosophila* non-muscle myosin II motor activity determines the rate of tissue folding

Claudia G Vasquez[1†‡], Sarah M Heissler[2†], Neil Billington[2], James R Sellers[2*], Adam C Martin[1*]

[1]Department of Biology, Massachusetts Institute of Technology, Cambridge, United States; [2]Laboratory of Molecular Physiology, National Heart, Lung, and Blood Institute, National Institutes of Health, Bethesda, United States

**Abstract** Non-muscle cell contractility is critical for tissues to adopt shape changes. Although, the non-muscle myosin II holoenzyme (myosin) is a molecular motor that powers contraction of actin cytoskeleton networks, recent studies have questioned the importance of myosin motor activity cell and tissue shape changes. Here, combining the biochemical analysis of enzymatic and motile properties for purified myosin mutants with in vivo measurements of apical constriction for the same mutants, we show that in vivo constriction rate scales with myosin motor activity. We show that so-called phosphomimetic mutants of the *Drosophila* regulatory light chain (RLC) do not mimic the phosphorylated RLC state in vitro. The defect in the myosin motor activity in these mutants is evident in developing *Drosophila* embryos where tissue recoil following laser ablation is decreased compared to wild-type tissue. Overall, our data highlights that myosin activity is required for rapid cell contraction and tissue folding in developing *Drosophila* embryos.

*For correspondence: sellersj@nhlbi.nih.gov (JRS); acmartin@mit.edu (ACM)

†These authors contributed equally to this work

Present address: ‡Chemical Engineering, Stanford University, Stanford, United States

Competing interests: The authors declare that no competing interests exist.

## Introduction

During the development of an organism, tissues are sculpted into different three-dimensional forms. Cell shape changes and cell movements drive these tissue-scale transformations. The molecular motor non-muscle myosin II (myosin) is thought to be important for many of these events; however, the underlying mechanism is not understood.

Myosin is a hexamer composed of two myosin heavy chains, two regulatory light chains (RLCs), and two essential light chains (ELCs) (*Figure 1A*). The motor domain at the N-terminus of the myosin heavy chain binds actin filaments in an ATP-dependent manner. The motor domain catalyzes the hydrolysis of ATP to power the translocation of actin filaments, a function referred to as motor activity. The light chains bind to the central neck domain of the myosin heavy chain and have structural and regulatory functions (*Heissler and Sellers, 2014*). The C-terminal tail of the myosin heavy chain associates with the tails of other myosin heavy chains and promotes the assembly into bipolar filaments. Work done using smooth muscle myosin and mammalian non-muscle myosin have demonstrated that phosphorylation of the RLC at conserved Serine and Threonine sites (*Figure 1B*, Serine-19 and Threonine-18) activates myosin motor activity, enhances the affinity of myosin for actin, and promotes myosin filament assembly (*Heissler and Sellers, 2016*). The bipolar myosin filaments promote the sliding of antiparallel actin filaments relative to one another resulting in contraction of an actin network. Myosin light chain kinase (MLCK), Rho-associated protein kinase (ROCK), and Citron kinase are known to activate myosin through direct phosphorylation of the RLC, primarily on Serine-19 (*Amano et al., 1996*; *Heissler and Sellers, 2014*; *Pearson et al., 1984*; *Yamashiro et al., 2003*). In *Drosophila*, the importance of myosin light chain phosphorylation has been shown through

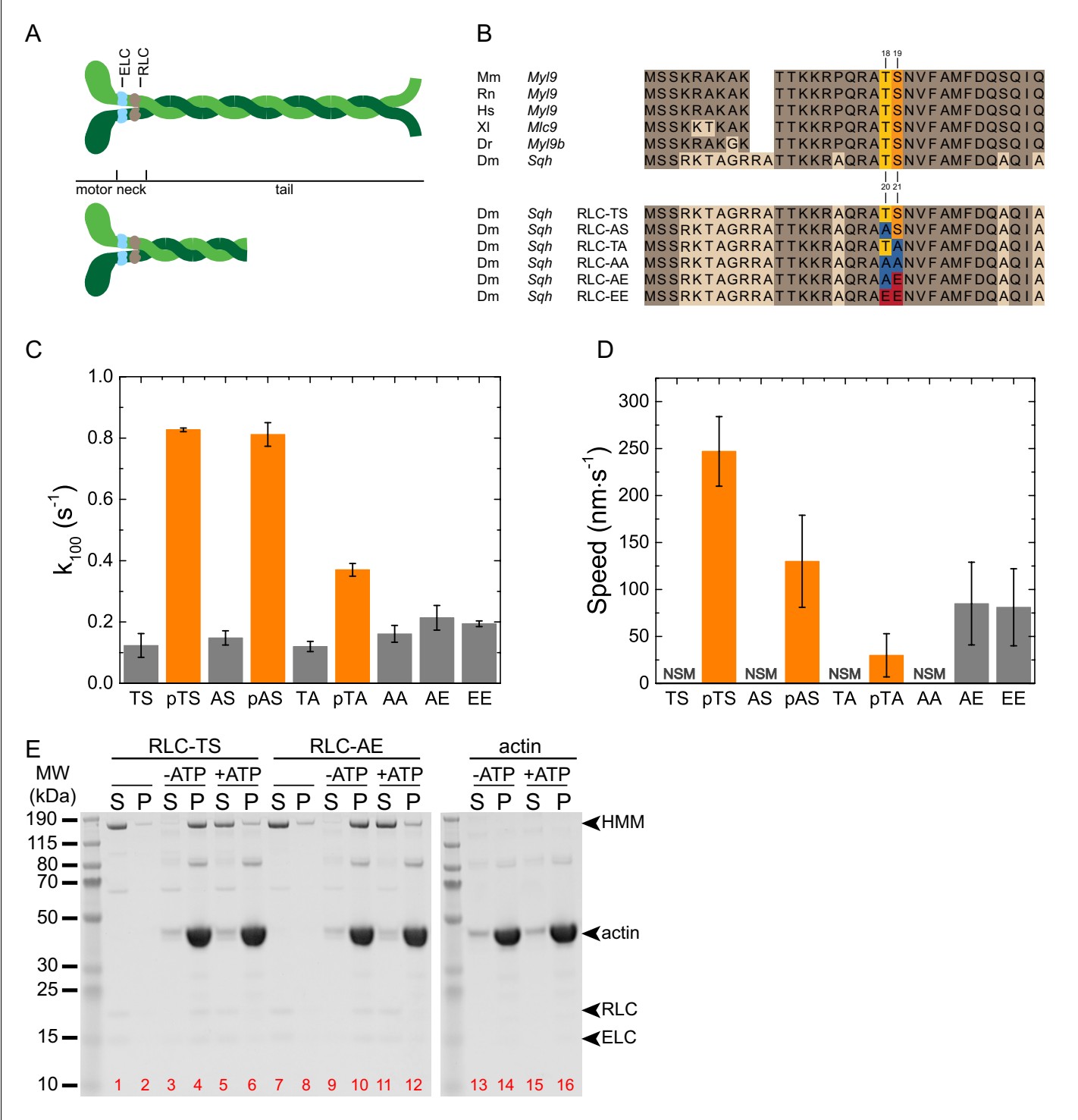

**Figure 1.** Biochemical characterization of RLC-TS and RLC mutants. (**A**) Domain organization of the myosin heavy chain and myosin fragments used to study the biochemical properties of myosin. The top panel shows the myosin hexamer composed of two myosin heavy chains (green), two ELCs (light blue) and two RLCs (gray). The myosin motor domain, the light chain binding neck and the tail domain of the heavy chain are indicated. The bottom panel shows the double headed HMM fragment. The rational for the different myosin fragments lies in the different biochemical properties: Full-length myosin forms filaments, sediments at high speed and can be used in the in vitro motility assay. The HMM fragment is soluble under physiological salt concentrations and suitable to study kinetic properties, the interaction with actin, and regulatory properties including RLC phosphorylation. (**B**) Multiple sequence alignment of RLCs from different model organisms and RLC mutants. (Top) Sequence alignment showing the high degree of conservation between RLCs from different model organisms. Identical amino acids are colored brown. The primary phosphorylation site of MLCK corresponds to

*Figure 1 continued on next page*

*Figure 1 continued*

Serine-21 (orange), the secondary phosphorylation site to Threonine-20 (yellow) of the *Drosophila* RLC. (Bottom) Alignment showing the mutant RLCs used in this study and the respective amino acid substitutions at the sites corresponding to Threonine-20 and Serine-21 of the *Drosophila* RLC. Alanine replacements are shown in blue, Glutamate replacements in red. Alanine replacement in RLC-AA is expected to mimic the unphosphorylated RLC state, Glutamate replacement in RLC-EE the di-phosphorylated RLC state. RLC-AE is expected to mimic the mono-phosphorylated RLC state that is independent from upstream regulation, whereas RLC-AS and RLC-TA are phosphorylatable and coupled to upstream signaling in vitro and in vivo. Abbreviations used: Mm: *Mus musculus*, *Myl9* (NP_742116.1): Rn: *Rattus norvegicus*, *Myl9* (XP_006235463.1); Hs: *Homo sapiens*, *Myl9* (CAG33124.1); Xl: *Xenopus laevis*, *Myl9* (NP_001087016.1); Dr: *Danio rerio*, *Myl9b* (NP_998377.1); Dm: *Drosophila melanogaster*, *Sqh* (NP_511057.1). (**C**) Actin-activated ATP hydrolysis rate of RLC-TS and RLC mutants. Gray bars indicate the ATP hydrolysis rate at an actin concentration of 100 µM in the absence of RLC phosphorylation. Orange bars indicate the ATP hydrolysis rate at an actin concentration of 100 µM after RLC phosphorylation with MLCK, also indicated by a lowercase p. Note that the myosin activity in these assays are even lower than shown since direct actin-mediated hydrolysis of ATP likely accounts for a substantial portion of the ATPase of the unphosphorylated and mutant samples. (**D**) In vitro motility assay using RLC-TS and RLC mutants. No significant movement was observed for RLC-TS in the absence of MLCK phosphorylation and RLC-AA. n = 47–227 tracked filaments per RLC mutant. The data were corrected for the stage drift. (**E**) Actin cosedimentation assays of RLC-TS and RLC-AE, supernatants (S) and pellets (P) from mixtures of myosin, actin, and ATP as indicated. In the absence of actin, RLC-TS and RLC-AE remain in the supernatant. Binding of RLC-TS and RLC-AE to actin in the absence of ATP results in pelleting of the actomyosin complex. The presence of ATP disassembles the actomyosin complex and myosin mostly remains in the supernatant. The sedimentation of actin is independent of ATP. Soluble HMM fragments were used in this assay since myosin filaments sediment under the assay conditions. Red numbers indicate lanes.

The following figure supplements are available for figure 1:

**Figure supplement 1.** Purification and characterization of wild-type and mutant *Drosophila* myosin (A) PageBlue-stained 4–12% Bis-Tris gel showing recombinant HMM (160 kDa) and full-length (228 kDa) RLC-TS or RLC mutants.

**Figure supplement 2.** Steady-state ATP-hydrolysis of RLC-TS (**A**) and RLC mutants (**B–F**).

genetic studies (*Jordan and Karess, 1997*; *Lee and Treisman, 2004*; *Mizuno et al., 2002*; *Ong et al., 2010*; *Winter et al., 2001*). However, it has not been biochemically demonstrated that *Drosophila* myosin motor activity and filament assembly is regulated by RLC phosphorylation or whether the extent of activation is similar to that of mammalian systems.

Recent evidence has suggested that myosin motor activity is not essential for actin network contraction in some cases, but that myosin's main role is to function as an actin filament crosslinker (*Ma et al., 2012*). Actin filament crosslinking could drive contraction in the absence of motor activity, if linked to actin network disassembly (*Sun et al., 2010*). It was suggested that myosin motor activity is not required for apical constriction in epithelial cells of the neural tube, but that instead actin depolymerization was required for epithelial folding (*Escuin et al., 2015*). In addition, it was recently suggested that apical constriction during *Drosophila* dorsal closure results from cell volume reduction as opposed to myosin motor activity (*Saias et al., 2015*). Thus, an important question is whether myosin motor activity has a role during apical constriction and tissue folding.

Common reagents used to test the necessity of myosin during cell and developmental processes are insufficient to determine the contribution of myosin motor activity. The ROCK inhibitor Y-27632 (*Uehata et al., 1997*) disrupts myosin filament assembly, motor activity, and prevents the motor domain from transitioning to the strong actin binding state by blocking RLC phosphorylation (*Sellers et al., 1982*). Furthermore, treatment with a ROCK inhibitor often disrupts myosin localization (*Hirose et al., 1998*; *Royou et al., 2002*; *Totsukawa et al., 2004*). The myosin inhibitor blebbistatin locks the motor domain in a weak actin binding state, a state with similar affinity to actin as that of myosin bound to an unphosphorylated RLC (*Kovács et al., 2004a*; *Ramamurthy et al., 2004*). While RLC mutants thought to mimic the activated state of myosin have been observed to enable some *rock* mutant flies to survive to adulthood (*Winter et al., 2001*), studies directly visualizing cell contractility determined that RLC mutants fail to suppress ROCK mutants (*Aranjuez et al., 2016*; *Kasza et al., 2014*; *Vasquez et al., 2014*) or ROCK inhibition (*Munjal et al., 2015*), questioning whether these mutants truly mimic the phosphorylated state. Overexpression of a myosin heavy chain mutant that lacks the motor domain demonstrated the importance of myosin for force generation during morphogenesis; however, this approach would disrupt both motor and crosslinking functions (*Franke et al., 2005*). Therefore, the relationship between myosin motor activity and cell and tissue shape change is unknown.

Defining the importance of myosin motor activity requires mutants with precisely characterized motor properties and quantitative analysis of cell and tissue shape changes. Previously, we made a series of *Drosophila* RLC mutants that substituted the phosphorylation sites with Alanine, as an unphosphorylatable residue or Glutamate, as a phosphomimetic (*Vasquez et al., 2014*). Here we show for the first time that *Drosophila* myosin motor activity and myosin filament assembly are regulated by phosphorylation of the RLC. Additionally, we show that so-called phosphomimetic alleles of the RLC do not fully mimic the behavior of myosins associated with phosphorylated RLCs. Rather, they have graded and measurable motor activity defects compared to myosins with phosphorylated RLC, providing us with an allelic series of myosin motor mutants. We find that decreased motor activity is associated with slower cellular contractions and slower folding of epithelial tissue. Our results show that the reduction in the rate of apical constriction scales with the extent of the deficit in myosin motor activity.

## Results

### RLC mutants have reduced motor activity

A prerequisite to systematically test the requirement of myosin motor activity during epithelial morphogenesis in *Drosophila* is to integrate in vitro and in vivo studies. Our approach included the preparation of wild-type myosin and RLC mutant myosins in sufficient amounts to biochemically characterize their kinetic, regulatory, and structural properties. This characterization of mutants with defined biochemical properties allows the precise interpretation of in vivo mutant phenotypes.

We first recombinantly overproduced and purified the heavy meromyosin (HMM) fragment and full-length *Drosophila* myosin in *Sf9* insect cells (*Figure 1A*, *Figure 1—figure supplement 1A*). Hereafter, the myosin holoenzyme with bound RLC is referred to as RLC-TS, where the phosphorylatable Threonine at position 20 and Serine at position 21 of the RLC are indicated (*Figure 1B*). Both sites are highly conserved between RLCs from higher eukaryotic myosins and their phosphorylation is expected to have an activating effect on the motor activity of *Drosophila* myosin (*Figure 1B*) (*Heissler and Manstein, 2013*, *2016*; *Jordan and Karess, 1997*).

To test whether phosphorylation of Threonine-20 and/or Serine-21 is a regulatory mechanism for *Drosophila* myosin, we examined the ability of MLCK to phosphorylate the RLC (*Figure 1—figure supplement 1B*) and used the actin-activated ATP hydrolysis rate under steady-state conditions as a readout for motor activity. The basal ATP hydrolysis rate ($k_{basal}$) in the absence of actin was ~0.01 $s^{-1}$ for RLC-TS, which is comparable to the rates of mammalian myosin isoforms (*Table 1*) (*Kim et al., 2005*; *Kovács et al., 2004b*). Actin marginally activated the ATP hydrolysis rate of RLC-TS in the absence of RLC phosphorylation. RLC phosphorylation potently activated myosin motor activity to a maximum ATP hydrolysis rate ($k_{cat}$) of 1.23 ± 0.07 $s^{-1}$ with an apparent actin affinity ($K_{app}$) of 38.9 ± 7.32 μM (*Table 2* and *Figure 1—figure supplement 2A*). This result demonstrates that *Drosophila* myosin is regulated by RLC phosphorylation and that the effect of RLC phosphorylation in *Drosophila* is quantitatively similar to myosins from higher eukaryotes (*Heissler and Manstein, 2013*).

**Table 1.** Comparison of the steady-state kinetic parameters between phosphorylated *Drosophila* and mammalian myosins.

| Parameter | *Dm* myosin II* | *Hs* myosin IIA[‡] | *Hs* myosin IIB [‡†] | *Mm* myosin IIC [‡†] |
|---|---|---|---|---|
| $k_{basal}$ ($s^{-1}$) | 0.01 ± 0.001 | 0.021 ± 0.003 | <0.004 | <0.004 |
| $k_{cat}$ ($s^{-1}$) | 1.23 ± 0.07 | 0.45 ± 0.03 | 0.17 ± 0.04 | 0.18 ± 0.04 |
| $K_{app}$ (μM) | 38.9 ± 7.32 | 9 ± 2 | 3.4 ± 1.8 | 4.2 ± 1.1 |
| $k_{cat}/K_{app}$ ($μM^{-1}·s^{-1}$) | ~0.03 | ~0.05 | ~0.05 | ~0.04 |

Abbreviations used: *Dm: Drosophila melanogaster; Hs: Homo sapiens; Mm: Mus musculus*.

*From this study.

[†]From *Kim et al. (2005)*.

[‡]From *Kovács et al. (2004a)*.

**Table 2.** Steady-state ATP hydrolysis rates of RLC-TS and RLC mutants.

| RLC | $k_{basal}$ (s$^{-1}$) | s.d. | $k_{100}$ (s$^{-1}$) | s.d. |
|---|---|---|---|---|
| RLC-TS | 0.013 | 0.007 | 0.12 | 0.03 |
| **RLC-TS** | **0.010** | **0.001** | **0.82** | **0.01** |
| RLC-AS | 0.024 | 0.005 | 0.15 | 0.02 |
| **RLC-AS** | **0.024** | **0.008** | **0.81** | **0.04** |
| RLC-TA | 0.009 | 0.002 | 0.12 | 0.01 |
| **RLC-TA** | **0.008** | **0.002** | **0.37** | **0.01** |
| RLC-AA | 0.014 | 0.006 | 0.16 | 0.03 |
| **RLC-AA** | **0.012** | **0.002** | **0.23** | **0.02** |
| RLC-AE | 0.010 | 0.002 | 0.21 | 0.03 |
| **RLC-AE** | **0.009** | **0.002** | **0.23** | **0.01** |
| RLC-EE | 0.011 | 0.007 | 0.19 | 0.01 |
| **RLC-EE** | **0.010** | **0.002** | **0.22** | **0.01** |

The actin-activated ATPase activity was measured at 25°C, as described under 'Materials and methods'. Treatment of RLC with MLCK is indicated by bold print. For comparison, $k_{basal}$ the steady-state ATPase activity in the absence of actin filaments, and $k_{100}$, the steady-state ATPase activity at 100 µM actin filaments, are listed.

Next, we generated the RLC mutants RLC-AS, RLC-TA, RLC-AA, RLC-AE, and RLC-EE (*Figure 1B*). Rationale for the design of these mutants, which are routinely used in the *Drosophila* system, is that they have been assumed to represent the unphosphorylated, mono- or di-phosphorylated RLC states (*Jordan and Karess, 1997*; *Kasza et al., 2014*; *Munjal et al., 2015*; *Rauskolb et al., 2014*; *Royou et al., 2002*; *Vasquez et al., 2014*; *Winter et al., 2001*). We tested the ability of MLCK to phosphorylate RLC mutants. We found that RLC-AS and RLC-TA can each be phosphorylated by MLCK at only one site, either Serine-21 or Threonine-20, respectively (*Figure 1—figure supplement 1C,D*). RLC-AA, RLC-AE, and RLC-EE were all unresponsive to MLCK treatment (*Figure 1—figure supplement 1E–G*). Thus, these mutants create a systematic series of myosins with motor activities that are either coupled (RLC-TA and RLC-AS) or uncoupled (RLC-AA, RLC-AE, and RLC-EE) from upstream signaling. This set of mutants also allowed us to individually determine the importance of phosphorylation of Threonine-20 and Serine-21 on myosin motor function.

Similar to RLC-TS, all RLC mutants have a low $k_{basal}$ of ~0.01–0.02 s$^{-1}$ (*Table 2*). Actin marginally activated the ATP hydrolysis rate of all RLC mutants to a similar extent (*Figure 1—figure supplement 2*). MLCK phosphorylation of RLC-AS increased its ATP hydrolysis rate to the same extent as that of the wild type RLC-TS (*Figure 1—figure supplement 2B*). In contrast, phosphorylation of RLC-TA activated the ATP hydrolysis rate to a reduced extent (*Figure 1C*, *Figure 1—figure supplement 2C*, and *Table 2*). The virtually identical steady-state ATP hydrolysis rates for RLC-TS and RLC-AS at 100 µM actin, but reduced activity of RLC-TA indicate that Serine-21 is the primary RLC phosphorylation site and that Threonine-20 is the secondary phosphorylation site. However, it is important to note that there is a slight reduction in the ATP hydrolysis rate of RLC-AS at lower actin concentrations (*Figure 1—figure supplement 2B*).

The almost linear dependence of the ATP hydrolysis rates of all RLC mutants on actin in the concentration range up to 100 µM (*Figure 1—figure supplement 2*) shows that both, the $k_{cat}$ and the $K_{app}$ are not experimentally accessible. The direct comparison of the ATP hydrolysis rates of RLC-TS and RLC mutants at 100 µM actin ($k_{100}$), the highest actin concentration used in the assay, shows that phosphorylation results in a ~6–7 fold activation for RLC-TS and RLC-AS versus a ~3 fold activation for RLC-TA, respectively (*Figure 1C* and *Table 2*). Surprisingly, RLC-AE and RLC-EE display a $k_{100}$ that was similar to that of RLC-AA and only 27–28% of the motor activity of phosphorylated RLC-TS (*Figure 1C*, *Figure 1—figure supplement 2D–F*, *Table 2*). This result demonstrates that neither RLC-AE nor RLC-EE successfully mimic RLC phosphorylation at Serine-21.

To test the functional activity of purified RLC-TS and RLC mutants, we performed an in vitro motility assay to measure the speed at which the motors translocate actin filaments across the surface of

a coverslip. No significant movement was detected for RLC-TS in the absence of phosphorylation, consistent with myosin being in an inactive, weak actin-binding state. Phosphorylation of RLC-TS by MLCK resulted in movement of actin filaments at ~250 nm s$^{-1}$, which is slightly higher than that of human myosin IIA (*Figure 1D* and *Table 3*) (*Cuda et al., 1997*). No obvious directed actin filament movement was observed for RLC-AA, consistent with this mutant mimicking the unphosphorylated RLC state. Phosphorylated RLC-AS moved actin filaments at 50% of the rate of phosphorylated RLC-TS and phosphorylated RLC-TA moved actin filaments at only 10% of the rate of phosphorylated RLC-TS. These results are consistent with Serine-21 being the primary phosphorylation site (*Figure 1D* and *Figure 1—figure supplement 2*). Both, RLC-AE and RLC-EE translocated actin filaments, demonstrating that these mutants exhibit some level of constitutive functional activity; however, translocation speeds of actin filaments by RLC-AE and RLC-EE were only about 30% of that of phosphorylated RLC-TS, indicating a functional deficit for these mutants (*Figure 1D*). Though neither unphosphorylated RLC-TS nor RLC-AA translocated actin filaments in the in vitro motility assay, both motors tethered actin filaments to the surface indicating that unphosphorylated *Drosophila* myosin still binds actin weakly as occurs with mammalian unphosphorylated myosins, suggesting that bipolar filaments containing these myosins could cross-link or tether actin filaments (*Cuda et al., 1997*).

To determine if the RLC-AE mutant drastically changes the myosin kinetic pathway, we performed an actin co-sedimentation assay to determine whether the mutant myosin binds to actin in a nucleotide-dependent manner. Actin is sedimented by high centrifugal forces regardless of the ATP concentration (lanes 13–16). Neither RLC-TS nor RLC-AE alone sediment without actin (lanes 1,2 and 7,8). When mixed with actin both RLC-TS and RLC-AE bind and sediment in the absence, but not the presence of ATP (lanes 3–6, 9–12). This suggests that RLC-TS and RLC-AE still cycle through strongly- and weakly-actin bound states during the kinetic cycle. Together, these results indicate that RLC-TS and RLC mutants likely use the same kinetic pathway, but that RLC-AE and RLC-EE do not fully recapitulate the behavior of myosin bearing wild type RLC. The RLC-AA mutant mimics the known functions of unphosphorylated RLC-TS in that it does not move actin in the in vitro motility assay, but does tether actin to the surface.

## RLC mutants can form bipolar filaments

Because myosin assembly into bipolar filaments is critical for both motor-dependent and motor-independent functions, such as translocation and crosslinking of actin filaments, we examined the ability of recombinant RLC-TS and RLC mutants to assemble into filaments. Electron micrographs show that RLC-TS and RLC mutants form small bipolar filaments, with a length of 312 ± 25 nm and a width of 7.4 ± 1.3 nm that consist of 12.8 ± 4.5 myosin molecules (*Figure 2*). The appearance of the filaments is similar to the rotary shadowed images of myosin purified from *Drosophila* cells and those formed by mammalian myosin IIC (*Table 3*) (*Billington et al., 2013*; *Kiehart and Feghali, 1986*).

We also revealed that full-length RLC-TS is in a two-state equilibrium, which involves a compact folded (10S) conformation and assembled filaments. The equilibrium is shifted from the filamentous form to the 10S conformation in the presence of ATP when the RLC is not phosphorylated (*Figure 2*). Similar to RLC-TS, all RLC mutants form filaments in the absence of ATP and disassemble into the 10S conformation in the presence of ATP (*Figure 2*). As occurs with mammalian myosin II paralogs,

**Table 3.** Comparison of *Drosophila* and mammalian myosin filament dimensions.

| Parameter | *Dm* myosin-II\*,[†] | *Hs* myosin IIA[‡] | *Hs* myosin IIB[‡] | *Hs* myosin IIC[‡] |
|---|---|---|---|---|
| Mean number of myosins per filament | 12.8 ± 4.5\* 14.9 ± 3[†] | 29 | 30 | 14 |
| Mean bare zone length (nm) | 194 ± 21\* 204 ± 19[†] | 167 ± 19 | 166 ± 16 | 219 ± 13 |
| Filament length (nm) | 312 ± 25\* | 301 ± 24 | 323 ± 24 | 293 ± 33 |
| Sliding velocity (nm·s$^{-1}$) | 247 ± 37 | 82[§] | n.d. | n.d. |

\*From this study.
[†]From *Kiehart and Feghali (1986)*.
[‡]From *Billington et al. (2013)*.
[§]From *Cuda et al. (1997)*.

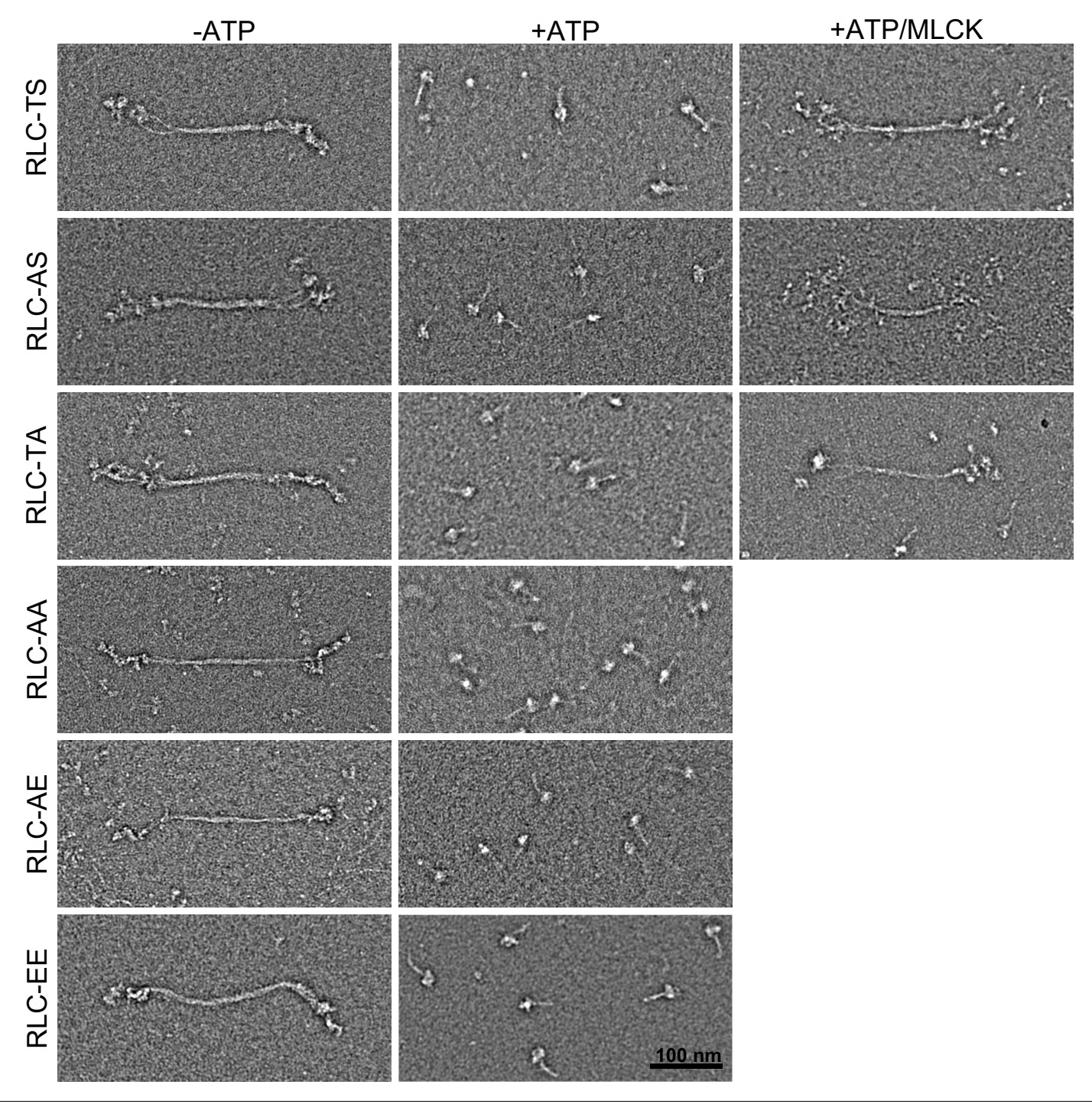

**Figure 2.** Appearance and assembly of RLC-TS and RLC mutants into filaments. RLC-TS and RLC mutants form predominantly filaments in the absence of ATP (−ATP) and disassemble into the 10S conformation in the presence of ATP (+ATP). Phosphorylation of Threonine-20 or Serine-21, indicated +ATP/MLCK, stabilize the filaments in the presence of ATP. Scale bar = 100 nm.

phosphorylated RLC-TS remains predominantly filamentous in the presence of ATP demonstrating a dual effect of phosphorylation; activation of the ATPase activity and stabilization of the filament structure (28). Phosphorylation also stabilizes filaments of RLC-AS and RLC-TA in the presence of ATP. If RLC-AE and RLC-EE were faithfully mimicking the phosphorylated RLC state, they should

also remain predominantly filamentous in ATP in vitro at these protein concentrations; instead, addition of ATP to RLC-AA, RLC-AE and RLC-EE results in depolymerization of the myosin filaments. These results demonstrate that the RLC-AS and the RLC-TA mutants exhibit phosphoregulation by upstream kinases similar to RLC-TS (*Figure 2*). The concentration of myosin in *Drosophila* cells is on the order of 0.5 µM, which favors the filamentous form and thus making it likely that all RLC mutants form filaments in vivo (*Kiehart and Feghali, 1986*; *Vasquez et al., 2014*). However, the assembly of RLC-AE and RLC-EE mutants in cells would not be governed by kinases. Thus, these RLC phospho-mutants are not only uncoupled from the control of signal transduction pathways that regulate myosin phosphorylation, but also display motor defects. These properties together prohibit RLC-AE and RLC-EE from being good models for phosphorylated myosin.

### *RLC-TA* and *RLC-AE* exhibit reduced tissue tension in vivo

We and others have constructed RLC mutants in *Drosophila* that affect the Threonine-20 and Serine-21 phosphorylation sites (*Jordan and Karess, 1997*; *Kasza et al., 2014*; *Vasquez et al., 2014*). Our mutant light chains are fused to GFP and expressed from the endogenous RLC promoter in a mutant background that reduces the expression of endogenous *RLC-TS*. We have quantified that there is 90% RLC mutant transgene expression and 10% endogenous *RLC-TS* in our experimental setup (*Vasquez et al., 2014*). To determine whether tissues expressing RLC mutants exhibit normal levels of apical myosin, we imaged pairs of embryos side-by-side, one *RLC-TS* and one RLC mutant, to compare apical myosin levels under identical imaging conditions (*Figure 3A*). Using this method, we demonstrated that the final apical myosin intensity in folding tissues was similar between *RLC-TS* and RLC mutant embryos (*Figure 3A,B*). These side-by-side comparisons of apical myosin levels, and our previous results where we showed that the *RLC-AA* mutant localized apically (23), demonstrate that neither full myosin motor activity nor targeted phosphorylation is required for apical myosin localization.

If the *RLC-AE* and *RLC-EE* mutants faithfully mimicked the phosphorylated RLC state in vivo, these so-called phosphomimetic mutants might be expected to result in increased tension relative to *RLC-TS* embryos. In our in vivo studies, we focused our analysis on the *RLC-AE* mutant, because the *RLC-EE* mutant often results in altered embryo shape, likely due to defects in oogenesis (*Vasquez et al., 2014*). Laser-cutting techniques have been used to infer the levels of tension in epithelial tissues (*Fernandez-Gonzalez et al., 2009*; *Fischer et al., 2014*; *Hutson et al., 2003*; *Ma et al., 2009*; *Mayer et al., 2010*). In response to a laser cut, the tissue adjacent to the incision will exhibit quick recoil away from the cut, followed by slower movement away from the cut site before the wound begins to heal and close. The initial rapid displacement of surrounding tissue, the initial recoil velocity, is proportional to the tension in the tissue just prior to the cut, assuming constant tissue viscosity (*Ma et al., 2009*) (see Materials and methods). Therefore, we compared the initial recoil velocity after tissue cutting in wild-type (*RLC-TS*) and RLC mutant embryos.

During ventral furrow formation in the developing *Drosophila* embryo, tension is highest along the anterior-posterior axis of the furrow (*Martin et al., 2010*). To infer tension in various RLC mutants, we made 8.5 µm long cuts across the width of the furrow at equivalently staged embryos and tracked the displacement of the wound edge along the anterior-posterior axis (*Figure 3C,D*). We calculated the initial recoil velocity (see Materials and methods) and found that tissues expressing either *RLC-TA* or *RLC-AE* display lower initial recoil velocities than *RLC-TS* tissues, suggesting that RLC mutant tissues generate less tension than *RLC-TS* tissues (*Figure 3D,E*). Note that this interpretation assumes that viscous drag is not different between tissues of mutant and *RLC-TS* embryos. Previously, altering the level of myosin phosphorylation was proposed to alter the viscoelastic properties of a tissue, as measured by a change in the exponent used when fitting the data to a power law (*Fischer et al., 2014*). We found that the mean power law exponent values for *RLC-TS*, *RLC-TA* and *RLC-AE* were 0.3, 0.3 and 0.4, respectively (*Figure 3—figure supplement 1A*). Differences between the power law exponents of *RLC-TS*, *RLC-TA*, and *RLC-AE* embryos were not significant (*Figure 3—figure supplement 1A*). Furthermore, we determined the time decay constants calculated from fitting recoil distance to a Kelvin-Voigt model, which are proportional to the ratio of the viscosity coefficient of the tissue to its stiffness. In agreement with power law exponents, differences between time decay constants of RLC-TS, RLC-TA, and RLC-AE embryos were not significant (*Figure 3—figure supplement 1B*). We suggest that the 3-fold decrease in initial recoil velocity in the *RLC-AE* mutant reflects decreased tension for the following reasons: (I) The power law exponent

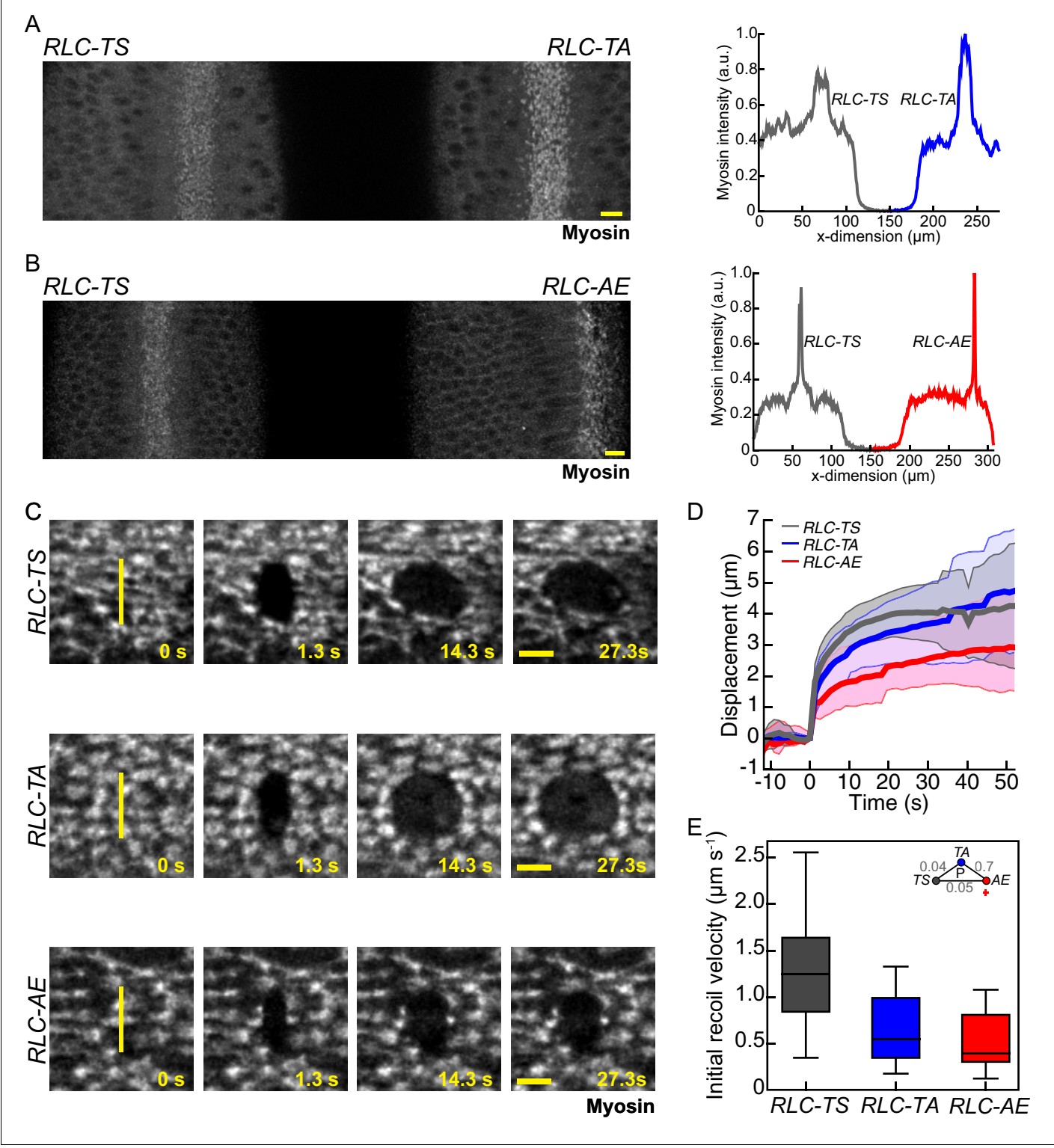

**Figure 3.** Myosin mutants with reduced motor activity exhibit reduced tissue recoil following laser ablation. (A–B) Myosin intensity in pair of embryos expressing *RLC-TS* and *RLC-TA* (A) or *RLC-TS* and *RLC-AE* (B) are comparable. The images represent embryos mounted side-by-side and imaged at the same time to compare myosin intensity under identical imaging conditions. Plots (right) show the peak mean intensity profile of myosin signal along a horizontal line spanning each embryo. Scale bars = 10 μm. (C) Initial wound recoil is less pronounced in tissues expressing RLC mutants than in *RLC-TS* tissue. Time-lapse images of embryos expressing indicated RLC mutant. Between time 0 s and 1.3 s an 8.5 μm incision in the tissue was made, indicated by the vertical yellow line. Scale bars = 5 μm. (D) Mean displacement of wound edge following tissue ablation (n = 15 *RLC-TS* embryos, *Figure 3 continued on next page*

*Figure 3 continued*

n = 11 *RLC-TA* embryos, n = 10 *RLC-AE* embryos). Shaded area is +/− one standard deviation. (E) Initial recoil velocity is higher in *RLC-TS* tissue. The initial recoil velocity was calculated by fitting the displacement to a Kelvin-Voigt model and taking the derivative of the fit at time zero. Central line in box plot is the median, the box edges are the 25th and 75th percentiles, and the whiskers represent the distribution minima and maxima (n = 14 *RLC-TS* embryos, n = 10 *RLC-TA* embryos, n = 9 *RLC-AE* embryos).

The following figure supplement is available for figure 3:

**Figure supplement 1.** *RLC-TS* and RLC mutant do not exhibit clear differences in viscoelastic properties.

and time decay constant are similar between *RLC-TS* and *RLC-AE*, suggesting similar viscoelastic properties; (II) Myosin levels are similar between *RLC-TS* and *RLC-AE*; (III) The *RLC-AE* mutant can still interact with actin, suggesting that there is not necessarily a difference in the amount of actomyosin present in the tissue; and (IV) *RLC-AE* exhibited a reduced maximal recoil distance (*Figure 3D*), which is also consistent with decreased tissue tension. We conclude that myosin motor activity is important to generate epithelial tension during ventral furrow formation.

## Apical constriction rate scales with the measured in vitro myosin motor activity

Understanding the contribution of the myosin motor to in vivo tissue folding requires having biochemically defined mutants or inhibitors that affect myosin motor activity. *Drosophila* myosin is insensitive to blebbistatin, a widely-used myosin inhibitor (*Heissler et al., 2015*; *Straight et al., 2003*), which has limited tests of in vivo myosin function to genetic disruptions or dominant negative approaches that do not specifically disrupt motor activity (*Franke et al., 2005*). Furthermore, inhibition of ROCK, a kinase that phosphorylates the RLC, completely disrupts myosin localization and does not distinguish between myosin motor and crosslinking functions (*Royou et al., 2002*). Although RLC-EE enables some *rock* mutant flies to survive to adulthood (*Winter et al., 2001*), apical constriction, tissue extension, and collective cell migration defects of *rock* mutants or ROCK inhibition are not suppressed by expressing the *RLC-EE* allele, suggesting that ROCK could have other important functional targets besides myosin or that RLC-EE does not mimic the phosphorylated state (*Aranjuez et al., 2016*; *Kasza et al., 2014*; *Munjal et al., 2015*; *Vasquez et al., 2014*). The in vivo effects of *RLC-AA* have already been reported (*Vasquez et al., 2014*). These mutants have defects in cellularization, which precedes tissue folding (*Vasquez et al., 2014*; *Xue and Sokac, 2016*). It is likely that the RLC-AA mutant is not able to oligomerize at all, in contrast to RLC-AE and RLC-EE, which form aggregates with heavy chain in vivo (*Vasquez et al., 2014*). In addition, because we demonstrated that RLC-AS, RLC-TA, and RLC-AE have variable deficits in motor activity that define an allelic series in vitro, these mutants were ideal to assess the in vivo roles of myosin motor activity during apical constriction.

To test the functional consequence of reducing myosin motor activity, we examined whether the *RLC-AE*, *RLC-TA*, and *RLC-AS* mutants constrict cells as well as *RLC-TS*. Because (I) there is great heterogeneity in the timing and rate of myosin accumulation between different cells and (II) the rate of myosin accumulation, as measured with apical RLC::GFP intensity, is correlated with the degree of constriction (*Xie and Martin, 2015*), the direct comparison of different RLC alleles required normalization for differences in myosin accumulation rates. To identify instances of rapid cellular constriction, we first smoothed signals for apical area and mean myosin intensity and calculated instantaneous rates for each time point. To normalize for differences in myosin accumulation rates, we identified times of rapid myosin intensity increase and selected time periods where myosin accumulated at a given rate (i.e. one to two standard deviations above the mean rate for *RLC-TS* cells). We calculated and compared constriction rates at these instances. Thus, we determined whether the same rate of apical myosin accumulation results in equivalent constriction rates in embryos expressing RLC mutants when compared to *RLC-TS*. We found that the mean constriction rate for cells in *RLC-AS* was 74% of that of *RLC-TS* cells (*Figure 4A*). This defect was consistent with the lower RLC-AS ATPase activity at intermediate actin concentrations and the lower actin gliding velocity of the RLC-AS mutant (*Figure 1D* and *Figure 1—figure supplement 2B*). We found that cells expressing *RLC-TA* or *RLC-AE* constricted with mean rates that were 60% and 47% of RLC-TS cells, respectively

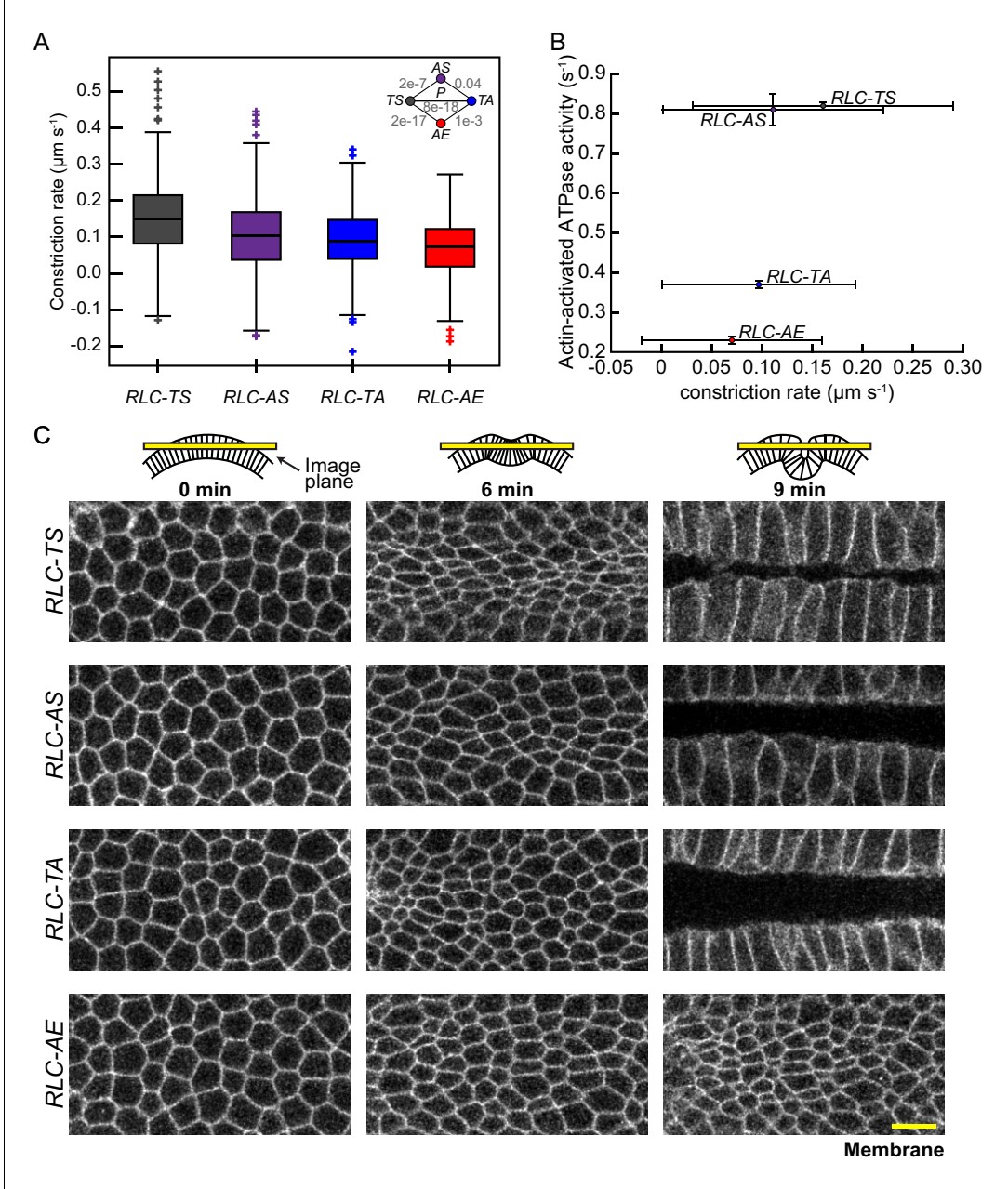

**Figure 4.** Apical constriction rate scales with measured in vitro myosin motor activity. (**A**) *RLC-AS*, *RLC-TA*, and *RLC-AE* cells generate weaker constrictions than *RLC-TS* cells. Central line in box plot is the median, the box edges are the 25th and 75th percentiles, and the whiskers represent the distribution minima and maxima, not considered outliers; outliers are plotted individually (n = 338 constriction instances identified from 138 cells from two embryos for *RLC-TS*; n = 311 constriction instances identified 152 cells from three embryos for *RLC-AS*; n = 630 constriction instances identified from 268 cells from three embryos for *RLC-TA*; n = 265 constriction instances identified from 187 cells from three embryos for *RLC-AE*). (**B**) Constriction rate corresponds with actin-activated ATPase activity. Bars represent the standard deviation of the measurements. (**C**) Tissue invagination and apical constriction is slower in the *RLC-TA* mutant and slowest in the *RLC-AE* mutant. Time-lapse images are representative tissues from embryos expressing the indicated RLC mutant and Membrane::Cherry (membrane). Schematics illustrate plane images were acquired in. Scale bars = 10 μm.

The following figure supplement is available for figure 4:

**Figure supplement 1.** Timing of tissue contraction scales with in vitro myosin motor activity.

(*Figure 4A*). The greater defect of the *RLC-TA* mutant, compared to *RLC-AS* was consistent with the Serine-21 site being the primary site responsible for myosin motor activation in *Drosophila* myosin, similar to mammalian myosins (*Bresnick et al., 1995*; *Heissler and Manstein, 2013*; *Ikebe, 1989*). Furthermore, on average, cells expressing *RLC-AE* constricted slower than those expressing *RLC-TA* (*Figure 4C*, *Figure 4—figure supplement 1*). Taking our in vivo and in vitro measurements of myosin activity, there is a relationship between myosin motor activity and constriction rate (*Figure 4B*). On the tissue-scale, the slower constriction is associated with a delay in tissue folding and sealing. The *RLC-AE* mutant, which had the greatest defect in motor activity, exhibited the slowest tissue folding behavior, with minimal folding occurring even 9 min after myosin first appeared (*Figure 4C*). The *RLC-AS* and *RLC-TA* mutants exhibited tissue folding at intermediate rates, folding over, but failing to seal after 9 min, a time in which *RLC-TS* is fully sealed (*Figure 4C*). Furthermore, the amount of time it takes the mean apical area to decrease to 50% of the starting apical area recapitulates the previously observed pattern: the *RLC-AE* mutant takes longest time (10.4 min) to reach 50% of the initial mean apical area, while the *RLC-AS* and *RLC-TA* mutants take an intermediate amounts of time (7.0 min and 7.4 min, respectively) relative to the wild-type *RLC-TS* (6.8 min) (*Figure 4—figure supplement 1*). Overall, our results show that the apical constriction rate and tissue folding rate scales with the myosin motor activity.

## Discussion

Whether myosin motor activity is required during tissue morphogenesis has not been clearly determined. Here, we combine in vitro analysis of the motor activity of RLC-TS and RLC mutants with in vivo characterization of force-generation by these motors at the cellular- and tissue-scales. A major advantage of our approach is that we have characterized the enzymatic properties of wild-type and mutant motors and quantified the in vivo effects of these motors within a single species, *Drosophila*. Such a systematic approach was essential to show that myosin motor activity is required for cellular contraction and to generate tension to fold epithelial tissues during *Drosophila* morphogenesis.

*Drosophila* has increasingly been used as a model system to understand the role of myosin in tissue morphogenesis. One significant advantage of *Drosophila* over mammalian systems is that *Drosophila* only has one gene encoding the myosin heavy chain, unlike the case with mammals where multiple myosin paralogs are expressed from three genes (*Golomb et al., 2004*; *Ketchum et al., 1990*; *Simons et al., 1991*). So far, it was assumed that *Drosophila* myosin is regulated identically to mammalian myosins by RLC phosphorylation although this cannot necessarily be assumed because striated muscle myosins from different species respond differently to RLC phosphorylation. For example, phosphorylation of the equivalent Serine residue in mammalian striated muscle myosin only slightly modulates the enzymatic and mechanical properties of the motor whereas in some invertebrates muscles, such as *Limulus* and *Tarantula*, phosphorylation of this site is an on/off switch as is found in mammalian non-muscle myosins (*Craig et al., 1987*; *Morgan et al., 1976*; *Sellers, 1981*). Here, we show for the first time that recombinant *Drosophila* myosin is regulated in an on/off manner by RLC phosphorylation. Phosphorylation of the RLC at Threonine-20 or Serine-21 activates myosin motor activity in addition to promoting the formation of bipolar filaments composed of 12.8 myosins under physiological conditions in vitro. Our study also shows that RLC phosphorylation at Threonine-20 results in different mechanoenzymatic properties than phosphorylation at Serine-21, in agreement with Serine-21 being the primary phosphorylation site (*Bresnick et al., 1995*). The similarity in regulatory properties between *Drosophila* myosin and vertebrate myosins qualifies *Drosophila* as an excellent model organism to study the underlying principles of myosin function and regulation in complex processes such as cell contraction and tissue folding.

RLC phosphorylation provides a means for spatial or temporal control of actomyosin contractility in cells. Mutants of the RLC phosphorylation sites in both *Drosophila* and mammalian cells have been used to test the role of RLC phosphorylation and its upstream regulation by signal transduction pathways in controlling myosin's activity and spatial distribution. Mutation of both Threonine-20 and Serine-21 to Alanines results in a myosin with properties similar to that of unphosphorylated mammalian myosin; the ATPase activity is low, there is no detectable movement of actin filaments in the in vitro motility assay and the myosin does not form filaments in the presence of ATP. In contrast and consistent with interpretations of past in vitro studies of smooth muscle myosin, we systematically show that substitution of phosphorylation sites of *Drosophila* RLC with negatively charged

residues (RLC-AE and RLC-EE) does not fully mimic the effects of phosphorylation of Serine-21 and Threonine-20 with respect to both the enzymatic and motile properties of the myosin (*Bresnick et al., 1995*; *Espinoza-Fonseca et al., 2014*; *Kamisoyama et al., 1994*; *Sweeney et al., 1994*). In addition, these mutants do not exhibit stable filament formation in vitro in the presence of ATP as occurs with phosphorylated wild type myosin. Furthermore these mutations would uncouple myosin activity from regulation by upstream kinases in vivo and in vitro (*Table 2*) (*Vicente-Manzanares et al., 2009*). The constitutive activity of these so-called phosphomimetic mutants was significantly lower than that of phosphorylated RLC-TS (*Figure 1C*), demonstrating that acidic residues are not sufficient to mimic the effect of phosphate moieties on Threonine-20 or Serine-21 of the RLC. This inadequacy is most likely due to substantial chemical differences between the phosphorylated amino acid side chain and the surrogate side chain, including size and formal charge at physiological pH. Reasons for the inability of negatively charged amino acids to recapitulate the phosphorylated RLC state include an allosteric communication pathway that involves the RLC, the ELC and the myosin motor domain and structurally requires the presence of a phosphate moiety (*Espinoza-Fonseca et al., 2014*). The results from the Glutamate replacements (RLC-AE and RLC-EE) in the current study together with results from recent molecular dynamic simulation study (*Espinoza-Fonseca et al., 2014*) lead us to hypothesize that mutant RLCs in which Aspartate is used as a replacement to mimic phosphorylation would also fail to faithfully mimic the phosphorylated RLC state (*Espinoza-Fonseca et al., 2014*; *Kamisoyama et al., 1994*; *Tang et al., 2011*). Taken together, our in vitro and in vivo data strongly suggests that caution should be used when interpreting the phenotype of so-called phosphomimetic RLC mutants in *Drosophila* and mammalian systems.

This study demonstrates that these RLC phosphomutants define an allelic series that can be used to test the function of myosin motor activity in morphogenetic processes in *Drosophila. Drosophila* myosin is insensitive to the pharmacological inhibitor blebbistatin (*Heissler et al., 2015*; *Straight et al., 2003*) and, until now, biochemically defined myosin mutants were unavailable. The *RLC-AS* mutant exhibited the smallest decrease in ATP hydrolysis rate at intermediate actin concentrations and also exhibited a marginal decrease in the in vivo constriction rate. The phosphorylation conditions of our in vitro biochemical experiments should result in phosphorylation primarily or exclusively at Serine-21. The results of our paper show that although Serine-21 is clearly the primary phosphorylation site, phosphorylation at Threonine-20 alone results in reduced actin-activated ATPase activity and a slower rate of actin filament sliding. Substituting an Alanine for Threonine-20 lowered the rate of actin filament sliding and had lower ATPase activity if assayed at low actin concentrations since the $K_{app}$ for this mutant was higher than for RLC-TS. This mutation (*RLC-AS*), when made in vivo also resulted in a reduced apical constriction rate, consistent with motor activity driving constriction. The *RLC-TA* mutant exhibited an intermediate defect in both in vitro and measured in vivo activities. Finally, the *RLC-AE* mutant had the most severe defect in the ATP hydrolysis rate and correspondingly had the most significant defect in constriction rate and tissue folding. Past work has suggested that *RLC-EE* can promote contraction (*Aldaz et al., 2013*; *Escudero et al., 2007*; *Kasza et al., 2014*). However, this result is only obtained when the *RLC-EE* is expressed in a background where wild-type *RLC-TS* is also present. Myosin molecules can form mixed filaments (i.e. hetero-oligomers) with other myosins (*Beach et al., 2014*). Thus, *RLC-EE* is likely forming mixed filaments with phosphorylated wild-type *RLC-TS* and recruiting the entire complex to an abnormal location to result in the excessive constriction. Importantly, *RLC-EE* is incapable of contraction without ROCK (*Kasza et al., 2014*; *Munjal et al., 2015*; *Vasquez et al., 2014*), suggesting that the wild-type phosphorylatable *RLC* is responsible for excessive cellular contraction in these experiments. Importantly, our in vitro motility assay results demonstrate that the phosphomutants RLC-AA, RLC-AE, and RLC-EE are clearly not mimicking phosphorylation, but can still bind F-actin. This is important because it suggests that these mutants can function as cross-linkers, if they oligomerize, but translocate actin more slowly when compared with the phosphorylated and unphosphorylated RLC-TS. In summary, motor activity scaled with in vivo contractile activity. Given this relationship, we propose that myosin motor activity determines the contraction rate during apical constriction and tissue folding. In the future, it will be important to determine whether other properties of myosin, like actin crosslinking, serve other roles during tissue morphogenesis.

## Materials and methods

### Recombinant overproduction and purification of *Drosophila* myosin and other proteins

cDNA constructs encoding N-terminally Halo-tagged full-length *Drosophila* myosin (amino acids 1–1971, accession number NP_001014553.1) and the HMM construct comprising amino acids 1–1363 were generated with standard cloning techniques. A polycistronic vector comprising the genes for *Sqh* (accession number NP_001284930.1) and *Mlc-c* (accession number NP_511049.1) was used for the expression of both light chain genes. Mutations of the cDNA at sites corresponding to amino acids Threonine-20 and Serine-21 of the RLC were made using the Quickchange XL kit (Agilent Technologies) to create the RLC mutants RLC-AA, RLC-AE, RLC-AS, RLC-EE, RLC-TA, and RLC-AS according to *Figure 1B*. Transposition, the generation of recombinant baculoviruses, protein production and purification were performed as described previously via Flag affinity chromatography (*Heissler et al., 2015*) with minor modifications. HMM was further purified to electrophoretic homogeneity with size exclusion chromatography on a HiLoad 16/600 Superdex 200 pg column (GE Healthcare). Full-length myosin was dialyzed in low salt buffer containing 10 mM MOPS pH 7.2, 25 mM NaCl, 0.1 mM EGTA, 3 mM NaN$_3$, 5 mM MgCl$_2$, and 2 mM DTT, concentrated by low speed centrifugation (4000xg, 15 min) and dissolved in 10 mM MOPS pH 7.2, 0.1 mM EGTA, 3 mM NaN$_3$, 500 mM NaCl, and 1 mM DTT. Actin was prepared from rabbit skeletal muscle acetone powder (Pel-Freez Biologicals) as described previously (*Heissler et al., 2015*). Rabbit smooth muscle MLCK and rat calmodulin were recombinantly overproduced in *Sf9* insect cells and *E.coli* and purified as described (*Wang et al., 2000*).

### HPLC-mass spectrometry

RLC-TS and RLC mutant HMMs (1–2.5 µM) were phosphorylated overnight in buffer containing 10 mM MOPS pH 7.3, 150 mM KCl, 5 mM MgCl$_2$, 0.2 mM CaCl$_2$, 0.1 mM EGTA, 1 mM DTT, 0.1 µM CaM, 0.2 mM ATP, 10 µg/mL MLCK, and 1x phosphatase inhibitor cocktail PhosSTOP (Roche, Indianapolis, IN) at 4°C. The conditions favor phosphorylation of both Threonine-20 and Serine-21 on RLC-TS whereas shorter incubation times favor phosphorylation of only Serine-21. MLCK was omitted for the unphosphorylated controls. A volume of 5–8 µl of the proteins was injected into a reverse phase HPLC (Agilent 1100 series HPLC, Agilent Technologies) with a Zorbax 300 SB-C18 (2.1×50 mm, 3.5 M, Agilent Technologies) and introduced into the mass spectrometer as described (*Apffel et al., 1995*; *Taggart et al., 2000*). Positive ion Electrospray Ionization (ESI) mass spectra for intact protein were obtained with an Agilent 6224 mass spectrometer equipped with an ESI interface and a time-of-flight (TOF) mass detector (Agilent Technologies). Mass spectra were analyzed and deconvoluted using MassHunter version B.06.00 (Agilent Technologies) software.

### ATPase activity and in vitro motility assay

Basal and actin-activated ATPase activities of HMM were determined using an NADH-linked assay as described previously (*Heissler et al., 2015*) at 25°C in buffer containing 10 mM MOPS (pH 7.0), 50 mM NaCl, 2 mM MgCl$_2$, 2 mM ATP, 0.15 mM EGTA, 40 units/mL lactate dehydrogenase, 200 units/mL pyruvate kinase, 1 mM phosphoenolpyruvate, 0.2 mM NADH and varying concentrations of filamentous actin (0–100 µM). Myosin HMM was phosphorylated at room temperature for a minimum of 30 min prior to the assay by the addition of 0.2 mM ATP, 0.2 mM CaCl$_2$, 0.1 µM CaM, and 10 µg/mL MLCK. To determine the kinetic constant $k_{cat}$ and $K_{app}$ experimental data sets were fit to the Michaelis-Menten equation using Origin 8.5 (OriginLab). The reported data represent the means and s.e. from at least four separate experiments. The in vitro motility assay was performed as described previously with full-length myosin in the presence or absence of MLCK phosphorylation (*Jana et al., 2009*).

### Actin-sedimentation assay

0.7 µM RLC-TS and RLC-AE HMM were incubated with 10 µM filamentous actin under nucleotide-free conditions or in the presence of 1 mM ATP in buffer containing 10 mM MOPS pH 7.3, 100 mM NaCl, 2 mM MgCl$_2$, 0.1 mM EGTA, 3 mM NaN$_3$ and 1 mM DTT for 30 min at room temperature. Samples only containing RLC-TS, RLC-AE or filamentous actin were used as a control to access the

sedimentation behavior. The samples were centrifuged (100,000xg, 30 min, 4°C) in a Beckman Optima MAX-XP. Supernatant and pellet fractions were separated and supplemented to an equivalent volume with loading buffer and 15 µl analyzed on a 4–12% Bis-Tris gel (Life Technologies). The gel was stained with PageBlue (Fermentas) and scanned on an Odyssey system (LiCor).

### Electron microscopy

Myosin was diluted to 100 nM in buffer containing 150 mM NaCl, 10 mM MOPS, 0.1 mM EGTA, 2 mM $MgCl_2$ (pH 7.0), (supplemented with 100 µM ATP where required). Samples were applied to the grid immediately after dilution. 3 µl of sample was applied to UV treated, carbon-coated copper grid and stained with 1% uranyl acetate. Micrographs were recorded on a JEOL 1200EX II microscope using an AMT XR-60 CCD camera, operating at 80.0 kV room temperature. Catalase crystals were used as a size calibration standard.

### Fly stocks and genetics

Fly stocks used in this study are listed in *Table 4*. *sqh* is the *Drosophila* gene for the myosin RLC. Briefly, we used *sqh-XX::GFP* alone and *sqh-XX::GFP* recombined with *Gap43::mCherry* (a plasma membrane) mutant stocks crossed into the *sqh[1]* hypomorph background, generated in *Vasquez et al. (2014)*. Germline clones were generated using the FLP-DFS technique by heat shocking mutant/*ovo[D]* larvae for 2 hr at 37°C for 3–4 d (*Chou and Perrimon, 1992*). Briefly, *sqh[1]* FRT/FM7; *sqh-XX::GFP/CyO* females were crossed to *ovo[D]* FRT/Y; *hsFlp* males, the resulting larvae were heat shocked, and *sqh[1]* FRT/*ovo[D]* FRT; *sqh-XX::GFP/hsFlp* females were crossed to OreR to collect embryos that resulted from *sqh[1]* mutant germline clones and also express *sqh-XX::GFP*. Throughout the manuscript, the notation distinguishes between recombinant RLC mutant protein and embryos expressing the RLC mutants by using normal (RLC-TS) versus italic (*RLC-TS*) type, respectively.

**Table 4.** Fly stocks.

| Genotype | Source | RRID for associated stocks or alleles* | |
|---|---|---|---|
| *Oregon-R-C (wild type)* | 2 | | RRID: FBst0000005 |
| *ovo[D1]FRT[101]/Y; hsFLP-38/hsFLP-38* | 2 | *ovoD1* | RRID: FBst0001813 |
| *sqh[1]FRT[101]/FM7; P{w+sqh-TS::GFP}attP1/CyO* | 1 | *sqh[1]*<br>*sqh-TS* | RRID: FBal0016066<br>RRID: FBal0298052 |
| *sqh[1]FRT[101]/FM7; P{w+sqh-AS::GFP}attP1/CyO* | 1 | *sqh[1]*<br>*sqh-AS* | RRID: FBal0016066<br>RRID: FBal0298055 |
| *sqh[1]FRT[101]/FM7; P{w+sqh-TA::GFP}attP1/CyO* | 1 | *sqh[1]*<br>*sqh-TA* | RRID: FBal0016066<br>RRID: FBal0298054 |
| *sqh[1]FRT[101]/FM7; P{w+sqh-AE::GFP}attP1/CyO* | 1 | *sqh[1]*<br>*sqh-AE* | RRID: FBal0016066<br>RRID: FBal0298057 |
| *sqh[1]FRT[101]/FM7; P{w+sqh-TS::GFP}attP1 P{w+Gap43::mCherry}attP40/CyO* | 1 | *sqh[1]*<br>*sqh-TS*<br>*Gap43* | RRID: FBal0016066<br>RRID: FBal0298052<br>RRID: FBtp0087760 |
| *sqh[1]FRT[101]/FM7; P{w+sqh-AS::GFP}attP1 P{w+Gap43::mCherry}attP40/CyO* | 3 | *sqh[1]*<br>*sqh-AS*<br>*Gap43* | RRID: FBal0016066<br>RRID: FBal0298055<br>RRID: FBtp0087760 |
| *sqh[1]FRT[101]/FM7; P{w+sqh-TA::GFP}attP1 P{w+Gap43::mCherry}attP40/CyO* | 1 | *sqh[1]*<br>*sqh-TA*<br>*Gap43* | RRID: FBal0016066<br>RRID: FBal0298054<br>RRID: FBtp0087760 |
| *sqh[1]FRT[101]/FM7; P{w+sqh-AE::GFP}attP1 P{w+Gap43::mCherry}attP40/CyO* | 1 | *sqh[1]*<br>*sqh-AE*<br>*Gap43* | RRID: FBal0016066<br>RRID: FBal0298057<br>RRID: FBtp0087760 |

*sqh-XX = sqh promoter and ORF with site-directed mutagenesis at Threonine-20 and Serine-21 as noted in **Figure 1B** (spaghetti squash, sqh is the Drosophila RLC gene name).*

*Gap43 = sqh promoter with N-terminal 20 amino acids of rat Gap43 gene which contains a myristoylation sequence (**Martin et al., 2010**).*

*FlyBase IDs (RRID: SCR_006549)

Sources: (1) From **Vasquez et al. (2014)**; (2) Bloomington Drosophila Stock Center; (3) This study;

## Time-lapse imaging

Embryos were dechorionated in 50% bleach, mounted ventral side upon a slide coated with 'embryo glue' (double-sided tape soaked in heptane). No. 1.5 coverslips were used as spacers and a No. 1 coverslip was used as a cover to create a chamber for the embryo. The chamber was filled with Halo-carbon 27 oil. All imaging occurred at room temperature on a Zeiss LSM 710 confocal microscope with a 40x/1.2 Apochromat water objective (*Figure 3C* and *Figure 4*) or 20X/1.0 Apochromat water objective (*Figure 3A and B*). Argon ion and 561 nm diode lasers were used for excitation. The band-pass selected for GFP was ~488–558 nm and for mCherry was ~573–696 nm. All images were acquired using Zen (Zeiss) software. For *Figure 3A* and *Figure 3B*, embryos of indicated genotype were mounted side-by-side on the same slide and imaged at the same time.

## Tissue-cutting

Ablations were performed using a Chameleon Ultra II femtosecond pulsed-IR laser (set at 800 nm and 24% intensity) attached to a Zeiss LSM710 with a 40X water immersion objective. To perform ablations, the laser was targeted to a region of interest (8.5 µm line). Image stacks were collected every 1.3 s.

## Image analysis

Images were processed using Fiji (RRID:SCR_002285), and MATLAB (Mathworks, RRID:SCR_001622).

### Myosin intensity profile of side-by-side embryos

In Fiji, a Gaussian filter ($\sigma = 1$) was applied to all images and a maximum projection was made of all z-slices. A mean intensity profile of the myosin signal along 33 µm of the length of the furrow over a region that extended across the width of both embryos was calculated. The time of the peak myosin intensity for each embryo was determined and the mean intensity profile for each was plotted in *Figure 3A and B*.

### Tissue-cutting analysis

In Fiji, for each ablation experiment, a Gaussian filter ($\sigma = 0.7$) was applied to all images and a maximum projection was made of both z-slices. Kymographs were taken along the center 4.25 µm of the laser incision. The image was converted to binary mode and a maximum projection was made from these kymographs. Images were imported into MATLAB, the built-in function bwmorph was used to remove outlier points. The distance between the wound edges was calculated for each time point. The displacement was this distance divided by two. The initial recoil velocity was calculated by fitting the displacement to a Kelvin-Voigt model of tissue displacement using the MATLAB build in function lsqcurvefit (*Equation 1*).

$$x(t) = \frac{T}{\varsigma}\left(1 - e^{\frac{\varsigma t}{\eta}}\right) \tag{1}$$

Where $t$ indicates time, $x(t)$ is the displacement of the wound edge, $T$ is the tensile force in the tissue prior to the cut, $\zeta$ is the stiffness of the spring, and $\eta$ is the viscous coefficient of the dashpot. Only fits with R-squared values greater than 50% were used in the subsequent calculations of initial recoil velocity. The initial recoil velocity ($v_0$) is described by the derivative of the displacement at time zero, which is the tension over the viscous coefficient of the dashpot (*Equation 2*). The time decay constant ($\tau$) is the viscous coefficient of the dashpot over the stiffness of the spring (*Equation 3*)

$$v_0 = \frac{dx(0)}{dt} = \frac{T}{\eta} \tag{2}$$

$$\tau = \frac{\eta}{\varsigma} \tag{3}$$

Displacement curves were fit to a power law function (*Equation 4*).

$$x(t) = \beta x^{\alpha} \tag{4}$$

The exponent ($\alpha$) of the power law gives a measurement of the viscoelastic properties of the cut tissues (*Fabry et al., 2001*; *Fischer et al., 2014*). Only fits with R-squared values greater than 50% were used.

## Constriction rate analysis

We used custom MATLAB software, EDGE (Embryo Development Geometry Explorer) (*Gelbart et al., 2012*) to segment images for quantification of apical area and myosin intensities. Segmented cell membranes were ~1 µm below the myosin signal. Embryos were aligned in time using the mean apical area signals of each embryo and choosing the time when the tissue begins to constrict. The myosin signal was pre-processed to remove cytoplasmic myosin by only using the maximum intensity z-projection for the two highest myosin intensity values that were greater than 2.5 standard deviations above the mean cytoplasmic myosin intensity. A Gaussian filter ($\sigma = 0.5$) was applied to this signal. To identify instances of rapid cellular constriction (*Figure 4A*), we first smoothed signals for apical area and mean myosin intensity and calculated instantaneous rates for each time point. Instances of rapid myosin intensity increases were identified as instances where the myosin accumulation rate was between one and two standard deviations above the mean rate for all control cells (*RLC-TS*). The constriction rate was calculated at these instances. To identify the amount of time it took the mean apical area of an embryo to decrease to 50% its starting area, we first aligned the mean apical areas of the embryos to each other by matching when the mean area starts to decrease. The initial apical area ($A_0$) is defined as the mean area 2 min before this rapid decrease, 0 min in *Figure 4—figure supplement 1A* for all representative embryos. We then determined the amount of time it took for the mean apical area to decrease to 50% of $A_0$.

## Statistics

All statistics were performed in MATLAB. *P*-values were calculated using the built-in ranksum function, which performs the Mann-Whitney U test.

## Download

EDGE source code is available at: https://github.com/mgelbart/embryo-development-geometry-explorer

Matlab code used for tissue-cutting and constriction rate analyses available at: https://github.com/clauvasq/myosin-motor

# Acknowledgements

We thank the WM Keck Microscopy Facility and Wendy Salmon for assistance with the use of the 2-photon laser and confocal microscope used in tissue-cutting experiments. We thank the Electron Microscopy Core Facility and the Biochemistry Core Facility of the National Heart, Lung, and Blood Institute for support, advice, and use of facilities. We thank Duck-Yeon Lee for his assistance with the mass spectrometry and analysis. This study was supported by the Intramural Research Program of the National Heart, Lung, and Blood Institute (SMH, NB and JRS), by R01GM105984 to AC Martin from the National Institute of General Medical Sciences, and by National Institutes of Health pre-doctoral training grant (T32GM007287).

# Additional information

### Funding

| Funder | Grant reference number | Author |
| --- | --- | --- |
| National Institute of General Medical Sciences | R01GM105984 | Adam C Martin |
| National Institute of General Medical Sciences | T32GM007287 | Claudia G Vasquez |

The funders had no role in study design, data collection and interpretation, or the decision to submit the work for publication.

### Author contributions
CGV, SMH, JRS, Conception and design, Acquisition of data, Analysis and interpretation of data, Drafting or revising the article; NB, Acquisition of data, Analysis and interpretation of data, Contributed unpublished essential data or reagents; ACM, Conception and design, Analysis and interpretation of data, Drafting or revising the article, Contributed unpublished essential data or reagents

### Author ORCIDs
Adam C Martin, http://orcid.org/0000-0001-8060-2607

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
