## [Decision Letter]

Thank you for submitting your article "*Drosophila* non-muscle myosin II motor activity determines the rate of tissue folding" for consideration by *eLife*. Your article has been reviewed by three peer reviewers, and the evaluation has been overseen by a Reviewing Editor and Marianne Bronner as the Senior Editor. The reviewers have opted to remain anonymous.

The reviewers have discussed the reviews with one another and the Reviewing Editor has drafted this decision to help you prepare a revised submission.

Summary:

In this manuscript, Vasquez et al. address the role that phosphorylation of the *Drosophila* myosin regulatory light chain (RLC) has on myosin-II motor activity, both in vitro and in vivo. By comparing results in these different assays, they are able to address directly the physiological correlation between biochemical activity and tissue phenotypes.

All three reviewers found this paper to be interesting and potentially suitable for *eLife*. They did all raise a number of issues that ought to be addressed, but these are mostly quite detailed and specific, so it is most efficient to list them fully below. The themes that emerged were a) the authors should describe the properties of the mutants in more detail; b) they should provide more detail/analysis of how their results do or don't correspond to previous work by others; and c) they need to improve/extend some aspects of their phenotypic analysis.

Overall, the feeling among the reviewers was that this would need little if any extra experimental work, as almost all the issues could be dealt with in the text.

1) There appear to be some differences in the biochemical properties of *Drosophila* myosin-II (e.g. Table 3 actin sliding velocity) and possibly the K_app_ (Table 2) compared to mammalian myosins. Are these differences in fact significant?

2) The actin co-sedimentation assays were difficult to follow in the text (Figure 1). Additional explanation of the assay, and which lanes to focus on in the Results, would be helpful.

3) Figure 1 shows that there are some differences in the in vitro motility of pAS mutants versus pTS and pTA. Although the results do support Serine-21 as the primary phosphorylation site, it is interesting that the pAS mutant has reduced but significant motility compared to pTS (and more than pTA). This is also seen in Figure 4. What role do the authors think this secondary phosphorylation site has on in vivo myosin functions? This could be helpful to add to the Discussion.

4) In a related comment, the authors show data for RLC-AS cell constriction in Figure 4 but do not show this mutant in Figure 4 – was the RLC-AS mutant tested in tissue invagination? If so, does this mutant have an intermediate effect between RLC-TS and RLC-TA (or is it more similar to RLC-TS)? In other words, is the secondary phosphorylation site truly negligible, or does this mutant show further graded activity of myosin in vivo?

5) On the last page of the Results, the authors state that "…the direct comparison of different RLC alleles required normalization for differences in myosin accumulation rates. Therefore, we identified instances with defined rates of apical myosin accumulation and determine the constriction rate at these instances." I found the description of this experiment in the Results to be a bit confusing, but it was much clearer in the Materials and methods.

6) In Figure 1, there seem to be some minor discrepancies between the actin-activated hydrolysis rates of different variants of RLC (Figure 1) and the motility assays (Figure 1). These discrepancies don't really fit with the idea of an allelic series of motor activities. The ATP hydrolysis rate of RLC-AA, RLC-AE and RLC-AE is about the same and barely above background (panel C). By contrast, there is no motility for RLC-AA and some for RLC-AE and EE (panel D)- why these discrepancies? In fact, the motility of AE and EE is better than pTA, which seems to contradict somewhat the results of the ATP hydrolysis assay.

7) The characterization of the phosphorylation states of the different RLC variants upon MLCK treatment (Figure 1—figure supplement 1) is only mentioned indirectly. For clarity, it would help if the results from the mass spectrometry would be reported more explicitly in the Results text.

8) I am not sure I understand what the conclusions are for the actin co-sedimentation assays. Both RLC-TS and RLC-AE HMM fragments behave the same, that is they bind actin in absence of ATP and this complex is disassembled in presence of ATP. Does that mean that they are good actin cross-linkers? Does that explain why, in vivo, TA and AE localize apically as well as TS? In general, there could be more discussion on the crosslinking activities of the RLC variants.

9) For the electron microscopy in Figure 2, are the full-length myosin treated with MLCK or only the RLC variants prior to myosin assembly? This was not clear from the Methods or the legend of Figure 2. If it is the full-length myosin, how sure are the authors that MLCK only phosphorylate RLC in the assay (and not MHC or ELC to stabilise filament formation for example)?

10) Figure 4: the authors did not look at the activity of RLC-AA in vivo; why is that? Would they expect AE or EE to have similar characteristics to AA, or in contrast, have higher activity compared to AA? in vitro, on one hand the ATP hydrolysis assays do not really discriminate between AA, AE and EE (Figure 1—figure – supplement 2 and Figure 1), but on the other hand, AE and EE do much better in the motility assay (Figure 1). More generally, do the authors think that the AE and EE variants have any function in vivo (which could be independent of motor activity)? Overall, it would be good to justify more clearly how mutants were selected for testing in vivo.

11) The authors show that RLC-AE and RLC-EE forms have lower actin-activated ATP hydrolysis rate and lower translocation speed of actin filaments than the phosphorylated forms of the wild-type. The authors also perform actin co-sedimentation assays and find that both wild-type RLC and RLC-AE co-sediment actin at similar levels. RLC-AS and RLC-TA also show lower motor activity than the wild-type, however, the results from the co-sedimentation assay are not shown. Are the non-phosphorylatable forms also able to co-sediment actin? If they are not, would it be possible that these proteins are not binding properly to actin, which would translate in a greater contractile defect?

12) The authors have extensively characterized different RLC phosphomutants, but there is one widely used phosphomutant, RLC-TE, for which no results are shown. Does this allele also have lower motor activity and lower constriction rate?

13) The authors find that phosphorylation stabilizes filaments of RLC-AS and RLC-TA. In contrast, addition of ATP to RLC-AA, RLC-AE and RLC-EE, results in depolymerisation of the myosin filaments. At what extent does this occur? If the stabilization of filaments by Myosin phosphorylation is similar for both RLC-AS and RLC-TA, does this mean that the formation of Myosin filaments is not dependent on the phosphorylation site?

14) The authors show that there are no significant differences in Myosin fluorescence intensity levels in the ventral furrow of embryos with the different Myosin forms. However, other groups have shown that the phosphomimetic form has increased levels at cell junctions (Kasza et al., 2014, Munjal et al., 2015) or increased Myosin pulse amplitude at the medioapical cortex of germband cells (Munjal et al., 2015). In the images shown here (Figure 3), it seems that the fluorescence levels are saturated in the case of RLC-TA and RLC-AE. The authors could compare images with lower intensity levels. Also, the RLC-AE embryo seems to be at a different stage than the others (the cells at the left of the ventral furrow look very much stretched). Because the authors are measuring the fluorescence intensity levels from maximum intensity projections (of how many slices?), I wonder whether changes in the apico-basal localization of Myosin could be interfering with the fluorescence intensity measurement. Showing the fluorescence intensity per cell at the apical cortex could support their observations.

15) The authors find that the initial recoil velocity after ablation is reduced in RLC-TA and RLC-AE. To show that the mechanical properties do not change, they do a power-law fit and find no differences between the phosphomutants and the wild-type. However, the maximal displacement looks different between RLC-TA and RLC-AE, with RLC-TA showing a similar maximal displacement to the wild-type. This would suggest that the stiffness could be changing in the RLC-AE. The authors could look at the time decay constant, which represents the ratio of viscosity to stiffness, and also analyse whether there are significant differences in the maximal displacement. This would shed light into whether the mechanical properties of the cells and tissue are changing in the phosphomutants, since this could also be an explanation of the lower cell and tissue constriction rate.

16) The authors conclude that Myosin motor activity determines the rate of cell and tissue constriction. However, the RLC-AS and RLC-TA have very different Actin-activated ATPase activity and the constriction rates are very similar (Figure 4), suggesting that the in vitro behaviour does not completely determine their in vivo activity. What is the rate of tissue invagination in RLC-AS mutants? Previously, the authors have shown that RLC-AE embryos do complete tissue invagination (Figure 3 in Vasquez et al., 2014), so it would be good to quantify the rate of tissue invagination, and to have an idea of the variability between embryos of the same genotype.

17) The authors conclude that RLC-AE and RLC-EE are not good models for phosphorylated myosin. Suprisingly, the alleles sqhEE and sqhDD (which would behave very similar to the sqhEE, as mentioned by the authors) have been shown to produce excessive cell contraction (Escudero et al., 2007; Aldaz et al., 2013), and to increase cell intercalation (Kasza et al., 2014). These results suggest that there is an activity of Myosin, which is indeed enhanced in these mutants.

[Editors' note: further revisions were requested prior to acceptance, as described below.]

Thank you for resubmitting your work entitled "*Drosophila* non-muscle myosin II motor activity determines the rate of tissue folding" for further consideration at *eLife*. Your revised article has been favorably evaluated by Marianne Bronner (Senior editor), a Reviewing editor, and three reviewers.

The reviewers all agree that the manuscript has been much improved, but before finally and formally accepting it, I'd appreciate it if you could consider the final few issues below. I will be happy to accept your judgement on the final decisions, but the reviewers did feel that these points might further improve an interesting and important paper.

1) It was felt to be useful to go ahead and add your response to point #10 to the text; this would give further context to the mutants shown in vivo.

2) More generally, it was noted that for some of the issues, you have responded in the rebuttal but not in the text. Will you look through these and consider bringing some of these points into the text to help the non-specialist reader. The reviewers feel that your paper successfully bridges several gaps in our knowledge of in vitro and in vivo systems, and as such will interest a broad audience, from biochemists to developmental biologists, so any effort to improve the accessibility of the manuscript would be appreciated.

As a specific example of this, the Introduction of the paper (third paragraph) poses the problem that crosslinking activity of myosin has been implicated in contractile activity independently of motor activity, but this is not really reexamined in light of your data in the Discussion (there is only a rather vague sentence: "In the future it will be important to determine whether other properties of myosin, like actin crosslinking, serve other roles during tissue morphogenesis"). Could you consider adding a section in the Discussion commenting on the cross-linking activities of myosin wild-type and mutant? (For example, see response to point 6: We included the in vitro motility assay in this work to demonstrate that the so-called "phosphomimetic" mutants RLC-AA, RLC-AE, and RLC-EE are clearly not mimicking phosphorylation, but can still bind F-actin. This is important because it suggests that these mutants can function as a cross-linker, but translocate actin more slowly when compared with the phosphorylated and unphosphorylated RLC-TS.)

Finally, regarding the laser ablation results, one of the reviewers still has a question that I'd encourage you to consider and decide whether it would be worth addressing in the text. You find that the initial recoil velocity (tension/viscosity) is decreased in both RLC-TA and RLC-AE mutants when compared to the wild-type, while the time decay constant (viscosity/stiffness) remains unchanged, supporting the idea that tension is lower in both these mutants. However, the maximal displacement (tension/stiffness) of RLC-TA is similar to the wild-type and the maximal displacement of RLC-AE is lower than the wild-type. How do you interpret this result?

---

## [Author Response]

*All three reviewers found this paper to be interesting and potentially suitable for eLife. They did all raise a number of issues that ought to be addressed, but these are mostly quite detailed and specific, so it is most efficient to list them fully below. The themes that emerged were a) the authors should describe the properties of the mutants in more detail; b) they should provide more detail/analysis of how their results do or don't correspond to previous work by others; and c) they need to improve/extend some aspects of their phenotypic analysis.*

*Overall, the feeling among the reviewers was that this would need little if any extra experimental work, as almost all the issues could be dealt with in the text.*

*1) There appear to be some differences in the biochemical properties of Drosophila myosin-II (e.g. Table 3 actin sliding velocity) and possibly the K_app_ (Table 2) compared to mammalian myosins. Are these differences in fact significant?*

There were some differences in the ionic strength and the actin concentrations of the various studies, which affect the K_app_, (which is an estimation of the affinity of myosin for actin in the presence of ATP). Differences in the ionic strength have less effect on the k_cat_. k_cat_ is probably more physiologically significant given that the actin concentration in cells is likely to be high. With regards to the in vitro motility assay, there are no published values for full length myosin IIB and myosin IIC. The values for myosin IIA came from a study from the Sellers lab in 1997 using tissue purified platelet myosin, which we now know is essentially pure myosin IIA. Although the buffer conditions of the current study and the Cuda et al. study are slightly different, it is clear that *Drosophila* myosin II is moving actin filaments faster than full length mammalian myosin IIA.

*2) The actin co-sedimentation assays were difficult to follow in the text (Figure 1). Additional explanation of the assay, and which lanes to focus on in the Results, would be helpful.*

We have added the text below to the Results and also added lane numbers to the figures.

“To determine if the RLC-AE mutant drastically changes the myosin kinetic pathway, we performed an actin co-sedimentation assay to determine whether the mutant myosin binds to actin in a nucleotide-dependent manner. […] The RLC-AA mutant does mimic known functions of unphosphorylated RLC-TS in that it does not move actin in the in vitro motility assay, but does tether actin to the surface.”

*3) Figure 1 shows that there are some differences in the in vitro motility of pAS mutants versus pTS and pTA. Although the results do support Serine-21 as the primary phosphorylation site, it is interesting that the pAS mutant has reduced but significant motility compared to pTS (and more than pTA). This is also seen in Figure 4. What role do the authors think this secondary phosphorylation site has on in vivo myosin functions? This could be helpful to add to the Discussion.*

The conditions of the in vitro motility assay phosphorylation reaction should result in only phosphorylation of Serine-21. In addition, Umemoto et al. JBC, 1989 showed that the velocity of di-phosphorylation of smooth muscle myosin at Threonine-18 and Serine-19 on actin was the same as that of myosin phosphorylated only at Serine-19, i.e. additional phosphorylation of the Threonine did not further increase velocity in contrast to its effects on ATPase activity where it does increase this rate. Thus, the difference in the motility rates of Figure 1 shown for pTS and pAS are likely due to the presence of the Alanine at residue 20. Please also note that the rate limiting step in the kinetic cycle for ATPase activity is likely to be phosphate release from an actomyosin:ADP-Pi complex, whereas the limiting step for in vitromotility is likely to be detachment of myosin from A.M which is essentially controlled by the rate of ADP release from actin-myosin-ADP. This is because the rate of ATP binding and subsequent dissociation of the actin:myosin complex is very fast. Thus, the differences seen in the effect of mutations on the in vitro motility and ATPase assays are not surprising.

*4) In a related comment, the authors show data for RLC-AS cell constriction in Figure 4 but do not show this mutant in Figure 4 – was the RLC-AS mutant tested in tissue invagination? If so, does this mutant have an intermediate effect between RLC-TS and RLC-TA (or is it more similar to RLC-TS)? In other words, is the secondary phosphorylation site truly negligible, or does this mutant show further graded activity of myosin in vivo?*

We added a new panel of images that show the RLC-AS mutant invagination behavior in Figure 4. We also added a new supplemental figure, Figure 4—figure supplement 1, that shows the mean apical area constriction of representative embryos (and averaged multiple embryos) over the time period analyzed in Figure 4. Based on the data shown in Figure 4, and Figure 4—figure supplement 1 we conclude that the RLC-AS mutant has less of an effect than the RLC-TA mutant. These results demonstrate that the Serine-21 site is the most important phosphorylation site in vivo, and further show that the effects of these mutants on biochemical activity are correlated with their in vivo function.

We added the following: “The *RLC-AE* mutant, which had the greatest defect in motor activity, exhibited the slowest tissue folding behavior, with minimal folding occurring even 9 minutes after myosin first appeared (Figure 4). […] Furthermore, the amount of time it takes the mean apical area to decrease to 50% of the starting apical area recapitulates the previously observed pattern: the *RLC-AE* mutanttakes longest time (10.4 minutes) to reach 50% of the initial mean apical area, while the *RLC-AS* and *RLC-TA* mutants take an intermediate amounts of time (7.0 minutes and 7.4 minutes, respectively) relative to the wild-type *RLC-TS* (6.8 minutes) (Figure 4—figure supplement 1).”

*5) On the last page of the Results, the authors state that "…the direct comparison of different RLC alleles required normalization for differences in myosin accumulation rates. Therefore, we identified instances with defined rates of apical myosin accumulation and determine the constriction rate at these instances." I found the description of this experiment in the Results to be a bit confusing, but it was much clearer in the Materials and methods.*

The reviewer made a good point and we have reworded this part accordingly:

“To identify instances of rapid cellular constriction, we first smoothed signals for apical area and mean myosin intensity and calculated instantaneous rates for each time point. To normalize for differences in myosin accumulate rates, we identified time of rapid myosin intensity increase and selected time periods where myosin accumulated at a given rate (i.e. one to two standard deviations above the mean rate for *RLC-TS* cells). We calculated and compared constriction rates at these instances.”

*6) In Figure 1, there seem to be some minor discrepancies between the actin-activated hydrolysis rates of different variants of RLC (Figure 1) and the motility assays (Figure 1). These discrepancies don't really fit with the idea of an allelic series of motor activities. The ATP hydrolysis rate of RLC-AA, RLC-AE and RLC-AE is about the same and barely above background (panel C). By contrast, there is no motility for RLC-AA and some for RLC-AE and EE (panel D)- why these discrepancies? In fact, the motility of AE and EE is better than pTA, which seems to contradict somewhat the results of the ATP hydrolysis assay.*

As the reviewer points out, there are some differences in relative activity levels between RLC mutants when comparing the ATPase and in vitromotility assays. This is likely because they are rate limited by different kinetic steps of myosin’s enzymatic cycle. The rate limiting step in the kinetic cycle for ATPase activity is likely to be phosphate release from the actomyosin:ADP-Pi complex. In contrast, the limiting step for the in vitro motility is likely to be detachment of myosin from actin:myosin which in turn is controlled by the rate of ADP release from actin:myosin:ADP since the rate of ATP binding and subsequent dissociation of the actin:myosin complex is very fast. Thus, the differences seen in the effect of mutations on the in vitro motility and ATPase assays are not surprising. Moreover, ADP release from myosin-II and other myosins is sensitive to the amount of load on the motor (Kovacs et al., *PNAS*, 2007; Laakso et al., Science, 2008). Because the in vivo load on the motor is unknown, how the velocity from the motility assay relates to in vivo motility is unknown. We included the in vitromotility assay in this work to demonstrate that the so-called “phosphomimetic” mutants RLC-AA, RLC-AE, and RLC-EE are clearly not mimicking phosphorylation, but can still bind F-actin. This is important because it suggests that these mutants can function as a cross-linker, but translocate actin more slowly when compared with the phosphorylated and unphosphorylated RLC-TS.

*7) The characterization of the phosphorylation states of the different RLC variants upon MLCK treatment (Figure 1—figure supplement 1) is only mentioned indirectly. For clarity, it would help if the results from the mass spectrometry would be reported more explicitly in the Results text.*

We agree with the reviewer, we now directly mention these results in the Results section:

“We tested the ability of MLCK to phosphorylate RLC mutants. We found that RLC-AS and RLC-TA can each only be phosphorylated by MLCK at one site, Serine-21 and Threonine-20 (Figure 1—figure supplement 1). […] This set of mutants also allowed us to individually determine the importance of phosphorylation of Threonine-20 and Serine-21 on myosin motor function.”

*8) I am not sure I understand what the conclusions are for the actin co-sedimentation assays. Both RLC-TS and RLC-AE HMM fragments behave the same, that is they bind actin in absence of ATP and this complex is disassembled in presence of ATP. Does that mean that they are good actin cross-linkers? Does that explain why, in vivo, TA and AE localize apically as well as TS? In general, there could be more discussion on the crosslinking activities of the RLC variants.*

These experiments were included to show that while the RLC-AE clearly has a major effect on the actin-activated ATPase activity and in vitro motility (Figure 1), it does not totally wreck the myosin kinetic cycle. All myosin II isoforms have the property that in the absence of ATP, myosin binds strongly to actin, whereas the presence of the nucleotide dramatically weakens the affinity for actin. The actin co-sedimentation assays directly show that RLC-TS and RLC-AE both bind and pellet with F-actin (lanes 4 and 10, respectively) to similar extents in the absence of ATP. In addition, the assay shows the responsiveness of both RLC-TS and RLC-AE to nucleotide state; we observe that myosin largely remains in the supernatant (lanes 5 and 11) and in the presence of ATP, which is what is expected for a proper cross-bridge cycle. The small fraction of myosin from both samples that sediments with actin is consistent with the weakening of the actomyosin affinity in the presence of ATP. We have rewritten this part of the Results to make our conclusions more clear.

*9) For the electron microscopy in Figure 2, are the full-length myosin treated with MLCK or only the RLC variants prior to myosin assembly? This was not clear from the Methods or the legend of Figure 2. If it is the full-length myosin, how sure are the authors that MLCK only phosphorylate RLC in the assay (and not MHC or ELC to stabilise filament formation for example)?*

MLCK is an incredibly specific enzyme for the phosphorylation of RLC. In 36 years of studying regulation of various myosin II isoforms by MLCK in the Sellers lab, we have never observed any indication of myosin heavy chain phosphorylation even in experiments where the phosphorylation of the secondary Threonine site was explored which required addition of 50-100 times as much kinase as was needed to phosphorylated the Serine residue only. We are not aware of any other substrate ever being described for MLCK. Furthermore, RLC-AA, RLC-AE, and RLC-EE were all unresponsive to MLCK treatment demonstrating that Threonine-20 and Serine-21 are the important residues for regulation (Figure 1—figure supplement 2).

*10) Figure 4: the authors did not look at the activity of RLC-AA in vivo; why is that? Would they expect AE or EE to have similar characteristics to AA, or in contrast, have higher activity compared to AA? in vitro, on one hand the ATP hydrolysis assays do not really discriminate between AA, AE and EE (Figure 1—figure supplement 2 and Figure 1), but on the other hand, AE and EE do much better in the motility assay (Figure 1). More generally, do the authors think that the AE and EE variants have any function in vivo (which could be independent of motor activity)? Overall, it would be good to justify more clearly how mutants were selected for testing in vivo.*

The in vivoeffects of *RLC-AA* have already been reported (Vasquez et al., JCB,2014). These mutants have defects in cellularization, which precedes tissue folding (Vasquez et al., JCB,2014; Xue and Sokac, JCB, 2016). It is likely that the RLC-AA mutant is not able to oligomerize at all, in contrast to RLC-AE and RLC-EE, which form aggregates with heavy chain in vivo (Vasquez et al.,JCB, 2014).

*11) The authors show that RLC-AE and RLC-EE forms have lower actin-activated ATP hydrolysis rate and lower translocation speed of actin filaments than the phosphorylated forms of the wild-type. The authors also perform actin co-sedimentation assays and find that both wild-type RLC and RLC-AE co-sediment actin at similar levels. RLC-AS and RLC-TA also show lower motor activity than the wild-type, however, the results from the co-sedimentation assay are not shown. Are the non-phosphorylatable forms also able to co-sediment actin? If they are not, would it be possible that these proteins are not binding properly to actin, which would translate in a greater contractile defect?*

We did not explore the actin-cosedimentation of the all the mutant forms of RLC, but the RLC-AE is a non-phosphorylatable form. We explored the requirement for phosphorylation in the in vitro motility assay. In numerous other studies from the Sellers lab we have shown that even though unphosphorylated smooth and nonmuscle myosins do not move actin filaments, they do tether actin filaments to the coverslip surface. This is almost certainly a result of the weak-actin binding affinity that was demonstrated in co-sedimentation assays described in Figure 1. Based on this we would expect all of the mutants to show low levels of co-sedimentation with actin in the presence of ATP and to be able to function as a weak actin tether if they form filaments inside the cell.

*12) The authors have extensively characterized different RLC phosphomutants, but there is one widely used phosphomutant, RLC-TE, for which no results are shown. Does this allele also have lower motor activity and lower constriction rate?*

The most commonly used phosphomutant is the RLC-EE mutant. In the examples given by the referees (point 17) where the phosphomimetic mutant results in excessive contraction (Escudero et al., Dev Cell, 2007; Aldaz et al., Nat Commun, 2013; Kasza et al., PNAS, 2014), the data mostly involves the *RLC-EE* or *RLC-DD* mutants. There is only one panel of one figure that shows a result involving the *RLC-TE* mutant. Therefore, we disagree that the *RLC-TE* mutant is widely used.

We already have addressed the function of the primary and secondary phosphorylation sites with Alanine mutants, where the interpretation is most clear. In addition, we tested the ability of glutamic acid to mimic phosphorylation, in mutants that did not contain phosphorylatable residues (*RLC-AE* and *RLC-EE*), where the interpretation is again most clear. Therefore, we respectfully disagree that biochemically characterizing the *RLC-TE* would change our conclusions or add to the manuscript.

*13) The authors find that phosphorylation stabilizes filaments of RLC-AS and RLC-TA. In contrast, addition of ATP to RLC-AA, RLC-AE and RLC-EE, results in depolymerisation of the myosin filaments. At what extent does this occur? If the stabilization of filaments by Myosin phosphorylation is similar for both RLC-AS and RLC-TA, does this mean that the formation of Myosin filaments is not dependent on the phosphorylation site?*

The reviewer is correct that the interpretation of Figure 2 is that phosphorylation of RLC-TA and RLC-AS stabilizes myosin filaments. Thus, filament formation requires phosphorylation, but is not dependent on the site phosphorylated.

*14) The authors show that there are no significant differences in Myosin fluorescence intensity levels in the ventral furrow of embryos with the different Myosin forms. However, other groups have shown that the phosphomimetic form has increased levels at cell junctions (Kasza et al., 2014, Munjal et al., 2015) or increased Myosin pulse amplitude at the medioapical cortex of germband cells (Munjal et al., 2015). In the images shown here (Figure 3), it seems that the fluorescence levels are saturated in the case of RLC-TA and RLC-AE. The authors could compare images with lower intensity levels. Also, the RLC-AE embryo seems to be at a different stage than the others (the cells at the left of the ventral furrow look very much stretched). Because the authors are measuring the fluorescence intensity levels from maximum intensity projections (of how many slices?), I wonder whether changes in the apico-basal localization of Myosin could be interfering with the fluorescence intensity measurement. Showing the fluorescence intensity per cell at the apical cortex could support their observations.*

None of the images we used to compare intensity values are saturated. In addition, the embryos were imaged side-by-side so that a direct intensity comparison could be made. The past studies that the reviewer mentioned (Kasza et al., PNAS, 2014; Munjal et al., Nature,2015) did not image embryos under the exact same imaging conditions, making it difficult to directly compare fluorescence intensity measurement. Nevertheless, we have done the experiment that the reviewer suggested: We have measure fluorescence intensity per cell at the apical cortex and it is the same when comparing *RLC-TS, RLC-TA*, and *RLC-AE* (Vasquez et al., JCB, 2014, Figure 5 and Figure supplement 4). We now provide the result where mutant and wild-type embryos are imaged under identical conditions and present that here as the most rigorous proof that the levels are not changed.

*15) The authors find that the initial recoil velocity after ablation is reduced in RLC-TA and RLC-AE. To show that the mechanical properties do not change, they do a power-law fit and find no differences between the phosphomutants and the wild-type. However, the maximal displacement looks different between RLC-TA and RLC-AE, with RLC-TA showing a similar maximal displacement to the wild-type. This would suggest that the stiffness could be changing in the RLC-AE. The authors could look at the time decay constant, which represents the ratio of viscosity to stiffness, and also analyse whether there are significant differences in the maximal displacement. This would shed light into whether the mechanical properties of the cells and tissue are changing in the phosphomutants, since this could also be an explanation of the lower cell and tissue constriction rate.*

The reviewer is correct in that there is a difference in the maximal displacement and also the initial recoil velocity. We did as the reviewer suggested and analyzed the time decay constant for fits to the Kelvin-Voigt model. We now include this analysis in Figure 3—figure supplement 1, which indicated there was not a significant difference between the *RLC-AE* and wild-type *RLC-TS* recoil curves. Together with the power law exponents we determined, our analysis suggests that the lower recoil velocity is due to lower tension and not a different stiffness of the tissue.

We state: “Differences between the power law exponents of *RLC-TS, RLC-TA*, and *RLC-AE* embryos were not significant (Figure 3—figure supplement 1). […] In agreement with power law exponents, differences between time decay constants of RLC-TS, RLC-TA, and RLC-AE embryos were not significant (Figure 3—figure supplement 1).

*16) The authors conclude that Myosin motor activity determines the rate of cell and tissue constriction. However, the RLC-AS and RLC-TA have very different Actin-activated ATPase activity and the constriction rates are very similar (Figure 4), suggesting that the in vitro behaviour does not completely determine their in vivo activity. What is the rate of tissue invagination in RLC-AS mutants? Previously, the authors have shown that RLC-AE embryos do complete tissue invagination (Figure 3 in Vasquez et al., 2014), so it would be good to quantify the rate of tissue invagination, and to have an idea of the variability between embryos of the same genotype.*

To illustrate the invagination of the different mutants, we have: 1) Added images of the *RLC-AS* mutant embryos to Figure 4, and 2) Provided quantification of the invagination/contraction for all mutants in Figure 4—figure supplement 1. The *RLC-AS* mutant embryos constrict only slightly less well than wild-type embryos and do not disrupt tissue folding as much as the *RLC-TA* mutant. Again, the effects we see on tissue folding are entirely consistent with the defects in ATPase activity observed in vitro. Note that we present data from 2-3 embryos per genotype, but have looked at many more embryos. What we present is representative of the folding behavior we have observed in many embryos per genotype.

*17) The authors conclude that RLC-AE and RLC-EE are not good models for phosphorylated myosin. Suprisingly, the alleles sqhEE and sqhDD (which would behave very similar to the sqhEE, as mentioned by the authors) have been shown to produce excessive cell contraction (Escudero et al., 2007; Aldaz et al., 2013), and to increase cell intercalation (Kasza et al., 2014). These results suggest that there is an activity of Myosin, which is indeed enhanced in these mutants.*

Some of the work the reviewer mentions involves overexpressing the *sqhEE* or *sqhDD* in a wild-type background (Escudero et al., Dev Cell, 2007; Aldaz et al., Nat Commun,2013). It is clear that the *sqhEE* mutant exhibits cortical localization in the absence of an upstream kinase (Vasquez et al., JCB, 2014; Kasza et al., PNAS, 2014; Munjal et al., Nature,2015). Functional myosin is an oligomer (i.e. bipolar filament consisting of 12 molecules in *Drosophila*). In addition, myosin molecules can form mixed filaments (i.e. hetero-oligomers) with other myosins (Beach et al., Curr Biol, 2014). Thus, overexpression of *sqhEE* is likely to form mixed filaments with phosphorylated wild-type *sqh* (for evidence for this, see Vasquez et al., JCB, 2014)and result in more wild-type myosins being polymerized into bipolar filaments. Because *sqhEE* is incapable of contraction without Rho Kinase (Vasquez et al., JCB, 2014; Kasza et al., PNAS,2014) it is the wild-type phosphorylatable *sqh* that is most likely promoting the excessive contractile activity.

In Kasza et al. the authors expressed *sqhEE* in a hypomorphic *sqh* background. There is still 10% levels of wild-type *sqh.* Indeed, making germline clones using a *sqh* null allele does not result in embryos (Vasquez et al. JCB, 2014), consistent with it lacking function. Kasza et al. did see excessive shortening of cell edges, but the subsequent elongation was defective (Kasza et al., PNAS, 2014). This elongation is driven by cortical actomyosin contractions in the apex (Collinet et al., NCB, 2015; Yu and Fernandez-Gonzalez, *eLife*, 2016). Thus, there is a germband elongation defect for *sqhEE* and it is likely due to a defect in myosin motor activity.

We state: “Past work has suggested that *RLC-EE* can promote contraction (Aldaz et al., 2013; Escudero et al., 2007; Kasza et al., 2014). […] Importantly, *RLC-EE* is incapable of contraction without ROCK (Kasza et al., 2014; Munjal et al., 2015; Vasquez et al., 2014), suggesting that the wild-type phosphorylatable *RLC* is responsible for excessive cellular contraction in these experiments.”

[Editors' note: further revisions were requested prior to acceptance, as described below.]

*The reviewers all agree that the manuscript has been much improved, but before finally and formally accepting it, I'd appreciate it if you could consider the final few issues below. I will be happy to accept your judgement on the final decisions, but the reviewers did feel that these points might further improve an interesting and important paper.*

*1) It was felt to be useful to go ahead and add your response to point #10 to the text; this would give further context to the mutants shown in vivo.*

Thanks for the suggestion, we added the following statement in the manuscript at the beginning of the “Apical constriction rate scales with the measured in vitro myosin motor activity” section: “The in vivo effects of RLC-AA have already been reported (Vasquez et al., JCB, 2014). These mutants have defects in cellularization, which precedes tissue folding (Vasquez et al., 2014; Xue and Sokac, 2016). It is likely that the RLC-AA mutant is not able to oligomerize at all, in contrast to RLC-AE and RLC-EE, which form aggregates with heavy chain in vivo (Vasquez et al., 2014).”

*2) More generally, it was noted that for some of the issues, you have responded in the rebuttal but not in the text. Will you look through these and consider bringing some of these points into the text to help the non-specialist reader. The reviewers feel that your paper successfully bridges several gaps in our knowledge of in vitro and in vivo systems, and as such will interest a broad audience, from biochemists to developmental biologists, so any effort to improve the accessibility of the manuscript would be appreciated.*

*As a specific example of this, the Introduction of the paper (third paragraph) poses the problem that crosslinking activity of myosin has been implicated in contractile activity independently of motor activity, but this is not really reexamined in light of your data in the Discussion (there is only a rather vague sentence: "In the future it will be important to determine whether other properties of myosin, like actin crosslinking, serve other roles during tissue morphogenesis"). Could you consider adding a section in the Discussion commenting on the cross-linking activities of myosin wild-type and mutant? (For example, see response to point 6: We included the in vitro motility assay in this work to demonstrate that the so-called "phosphomimetic" mutants RLC-AA, RLC-AE, and RLC-EE are clearly not mimicking phosphorylation, but can still bind F-actin. This is important because it suggests that these mutants can function as a cross-linker, but translocate actin more slowly when compared with the phosphorylated and unphosphorylated RLC-TS.)*

We have gone through our rebuttal and think that all relevant points addressed in the rebuttal are in the main text of the manuscript. We added one point in the Discussion regarding actin crosslinking: “Importantly, our in vitro motility assay results demonstrate that the phosphomutants RLC-AA, RLC-AE, and RLC-EE are clearly not mimicking phosphorylation, but can still bind F-actin. This is important because it suggests that these mutants can function as cross-linkers, if they oligomerize, but translocate actin more slowly when compared with the phosphorylated and unphosphorylated RLC-TS.”

*Finally, regarding the laser ablation results, one of the reviewers still has a question that I'd encourage you to consider and decide whether it would be worth addressing in the text. You find that the initial recoil velocity (tension/viscosity) is decreased in both RLC-TA and RLC-AE mutants when compared to the wild-type, while the time decay constant (viscosity/stiffness) remains unchanged, supporting the idea that tension is lower in both these mutants. However, the maximal displacement (tension/stiffness) of RLC-TA is similar to the wild-type and the maximal displacement of RLC-AE is lower than the wild-type. How do you interpret this result?*

The reduction in maximal recoil displacement for RLC-AE is consistent with this mutant having the greatest reduction in tissue tension. We have added the following sentence to indicate our interpretation, “RLC-AE exhibited a reduced maximal recoil distance (Figure 3), which is also consistent with decreased tissue tension”.